# Deletion of a *Csf1r* enhancer selectively impacts CSF1R expression and development of tissue macrophage populations

Rocío Rojo [1,2], Anna Raper [1], Derya D. Ozdemir[1], Lucas Lefevre [1], Kathleen Grabert [1,3], Evi Wollscheid-Lengeling[1], Barry Bradford [1], Melanie Caruso [1], Iveta Gazova [1], Alejandra Sánchez[1], Zofia M. Lisowski [1], Joana Alves [1], Irene Molina-Gonzalez [4], Hayk Davtyan [5], Rebecca J. Lodge[6], James D. Glover[1], Robert Wallace[7], David A.D. Munro[8], Eyal David[9], Ido Amit [9], Véronique E. Miron[4], Josef Priller[8], Stephen J. Jenkins [6], Giles E. Hardingham[8,10], Mathew Blurton-Jones[5], Neil A. Mabbott [1], Kim M. Summers [11], Peter Hohenstein[1,12], David A. Hume[11,14] & Clare Pridans [6,13,14]

The proliferation, differentiation and survival of mononuclear phagocytes depend on signals from the receptor for macrophage colony-stimulating factor, CSF1R. The mammalian *Csf1r* locus contains a highly conserved super-enhancer, the *fms*-intronic regulatory element (FIRE). Here we show that genomic deletion of FIRE in mice selectively impacts CSF1R expression and tissue macrophage development in specific tissues. Deletion of FIRE ablates macrophage development from murine embryonic stem cells. $Csf1r^{\Delta FIRE/\Delta FIRE}$ mice lack macrophages in the embryo, brain microglia and resident macrophages in the skin, kidney, heart and peritoneum. The homeostasis of other macrophage populations and monocytes is unaffected, but monocytes and their progenitors in bone marrow lack surface CSF1R. Finally, $Csf1r^{\Delta FIRE/\Delta FIRE}$ mice are healthy and fertile without the growth, neurological or developmental abnormalities reported in $Csf1r^{-/-}$ rodents. $Csf1r^{\Delta FIRE/\Delta FIRE}$ mice thus provide a model to explore the homeostatic, physiological and immunological functions of tissue-specific macrophage populations in adult animals.

[1] The Roslin Institute & Royal (Dick) School of Veterinary Studies, University of Edinburgh, Easter Bush, Midlothian EH25 9RG, UK. [2] Tecnologico de Monterrey, Escuela de Medicina y Ciencias de la Salud, Av. Ignacio Morones Prieto 3000 Pte, Col. Los Doctores, C.P. 64710 Monterrey, N.L., Mexico. [3] Department of Environmental Medicine, Toxicology Unit, Karolinska Institutet, Box 210SE-171 77 Stockholm, Sweden. [4] The MRC University of Edinburgh Centre for Reproductive Health, The Queen's Medical Research Institute, Edinburgh BioQuarter, 47 Little France Crescent, Edinburgh EH16 4TJ, UK. [5] Department of Neurobiology and Behavior, University of California Irvine, 3014 Gross Hall 845 Health Sciences Rd, Irvine, CA 92697-1705, USA. [6] University of Edinburgh Centre for Inflammation Research, The Queen's Medical Research Institute, Edinburgh BioQuarter, 47 Little France Crescent, Edinburgh EH16 4TJ, UK. [7] The Department of Orthopedic Surgery, University of Edinburgh, Chancellor's Building, Edinburgh BioQuarter, 49 Little France Crescent, Edinburgh EH16 4SB, UK. [8] UK Dementia Research Institute, The University of Edinburgh, Chancellor's Building, Edinburgh BioQuarter, 49 Little France Crescent, Edinburgh EH16 4SB, UK. [9] Department of Immunology, Weizmann Institute of Science, 234 Herzl St., Rehovot 7610001, Israel. [10] Centre for Discovery Brain Sciences, University of Edinburgh, Hugh Robson Building, 15 George Square, Edinburgh EH8 9XD, UK. [11] Mater Research Institute-University of Queensland, Translational Research Institute, Woolloongabba, QLD 4102, Australia. [12] Leiden University Medical Center, P.O. Box 96002300 RC Leiden, The Netherlands. [13] Simons Initiative for the Developing Brain, Centre for Discovery Brain Sciences, University of Edinburgh, Hugh Robson Building, George Square, Edinburgh EH8 9XD, UK. [14]These authors jointly supervised this work: David A. Hume, Clare Pridans. Correspondence and requests for materials should be addressed to D.A.H. (email: David.Hume@uq.edu.au) or to C.P. (email: clare.pridans@ed.ac.uk)

The proliferation, differentiation, and survival of vertebrate macrophages is controlled by signals from the macrophage colony-stimulating factor (CSF1) receptor (CSF1R), encoded by the *c-fms* protooncogene[1], now known as *Csf1r*. The function of the receptor and its two ligands, CSF1 and interleukin 34 (IL34), in macrophage differentiation is conserved in all amniotes (i.e. reptiles, birds, and humans)[2,3]. Deletion of the *Csf1r* locus in the mouse or rat germ line produces a global deficiency in most tissue macrophage populations[4,5]. Mutant animals are osteopetrotic (associated with loss of osteoclasts) and exhibit severe postnatal growth retardation and multiple pleiotropic impacts on development. Tissue macrophages remain dependent upon CSF1R in adult mice and can be depleted by treatment with an inhibitory anti-CSF1R antibody[6] or treatment with orally available inhibitors of CSF1R kinase activity[7]. In humans, dominant-negative mutations in the tyrosine kinase domain are associated with an adult-onset neurodegenerative disease[8].

The expression of *Csf1r* mRNA is restricted to myeloid cells. During embryonic development, *Csf1r* mRNA is expressed in the earliest macrophages identifiable in the yolk sac and transcription of this gene in pluripotent bone marrow (BM) progenitors is the hallmark of commitment to the monocyte–macrophage lineage. Accordingly, the molecular basis for myeloid-restricted transcription regulation of *Csf1r* has been studied in considerable detail (reviewed in ref. [9]). The second intron, downstream of the first coding exon, contains a conserved 337bp sequence known as the *fms*-intronic regulatory element (FIRE), identified as a super-enhancer in genome-wide analysis of chromatin in mouse macrophages. The mouse FIRE sequence contains binding sites for numerous macrophage-expressed transcription factors[9]. The combination of a 3.5 kb *Csf1r* promoter and intron 2, containing FIRE, is sufficient to direct reproducible transgenic reporter gene expression in progenitors, monocytes, granulocytes, classical dendritic cells, and tissue macrophages that also express *Csf1r* mRNA[10,11]. Removal of FIRE from the reporter construct abolishes expression[10]. The same *Csf1r* construct was used to drive conditional Cre-recombinase in lineage-trace studies that dissected the role of the yolk sac in macrophage development[12]. In mammals, the FIRE sequence is more highly conserved than any of the exons of the *Csf1r* gene[13]. A FIRE sequence is present in the same relative intronic location of the *Csf1r* locus in all reptile and bird species, and a core element required for enhancer activity[14] is perfectly conserved[13]. The chicken *CSF1R* promoter and FIRE sequences are also sufficient to direct macrophage-specific expression in chicken transgenic lines and to highlight the appearance of the earliest phagocyte populations in the yolk sac[15].

There have been relatively few studies of the function of macrophage-specific enhancers in their genomic context. A highly conserved upstream regulatory element that is present in the *Spi1* locus (which encodes the macrophage-specific transcription factor PU.1) is required for regulated expression in myeloid cells. Deletion of this element in the germline produces a failure of myeloid differentiation and development of acute myeloid leukemia[16]. Deletion of a super-enhancer within the locus of the orphan nuclear receptor transcription factor (*Nr4a1*) reveals its functional specificity, as only the LY6C[lo] monocytes are lost, whilst the sensitivity of other monocyte and macrophage subsets to pro-inflammatory stimuli is preserved[17].

We aimed to determine the function of FIRE by deleting it from the mouse germ line. If FIRE is essential for *Csf1r* transcription, genomic deletion should partly or completely phenocopy the loss of tissue macrophages observed in *Csf1r*$^{-/-}$ mice. Here we report that deletion of FIRE has an organ-specific effect on *Csf1r* expression and macrophage differentiation. Despite the lack of brain microglia and resident macrophages in skin, heart,

kidney, and peritoneum, *Csf1r*$^{\Delta FIRE/\Delta FIRE}$ mice are healthy and fertile. Aside from highlighting the likely function of other regulatory elements in the *Csf1r* locus, these mice provide a model to explore tissue-specific macrophage transcriptional regulation and function.

## Results

**In vitro validation of CRISPRs targeting the FIRE sequence.** Guide RNAs (gRNAs) designed to delete FIRE (Fig. 1a) were first validated in the RAW 264.7 macrophage cell line and in E14 mouse embryonic stem cells (ESC). The macrophage cell line expresses *Csf1r* mRNA and was used in all of the transfection studies that previously led to the characterization of the role of FIRE[14]. Both cell types were co-transfected with pairs of Cas9-2A-EGFP constructs each expressing single gRNAs (US1+DS1 or DS2), and pools of EGFP$^+$ cells were screened via PCR to detect the deletion of FIRE mediated by the CRISPR/Cas9 system. Both pairs of gRNAs produced deletions in E14 ESC and RAW 264.7 cells (Supplementary Fig. 1a). The deletion of FIRE and the absence of mismatches in the remaining *Csf1r* sequence in E14 ESC were confirmed by Sanger sequencing (Supplementary Fig. 1b). In RAW 264.7 macrophage cells, individual *Csf1r*$^{\Delta FIRE/\Delta FIRE}$ clones carried slightly different deletions of the FIRE sequence (Supplementary Fig. 1c). RAW 264.7 cells are not CSF1R-dependent and most of the cellular CSF1R protein is retained in the Golgi apparatus. We therefore analyzed cell-associated CSF1R by flow cytometry in permeabilized cells. The deletion of FIRE completely ablates detectable intracellular CSF1R (Fig. 1b). There is no impact of the FIRE mutation on cell morphology (Fig. 1c) or phagocytic capacity of cells (Fig. 1d). Neither the frequency nor the median fluorescence intensity (MFI) of the macrophage surface marker F4/80 distinguishes between *Csf1r*$^{+/+}$ and *Csf1r*$^{\Delta FIRE/\Delta FIRE}$ clones (Fig. 1e). We conclude that FIRE is required specifically for the expression of CSF1R in the RAW 264.7 macrophage cell line.

E14 mouse ESC can be cultured in the absence of feeders and form embryoid bodies (EB)[18] with the potential to generate cells belonging to all the primary embryonic germ layers (i.e. endoderm, mesoderm, and ectoderm). Macrophage differentiation from EB was induced by supplementing culture medium with murine IL3 (mIL3) and recombinant human CSF1 (rhCSF1). *Csf1r*$^{\Delta FIRE/\Delta FIRE}$ mouse ESC generated using the gRNA pair US1+DS2 are able to produce EB that are indistinguishable from the controls (Fig. 1f, Day 0). By day 7, they give rise to cells that attach to culture plates, indicating that the deletion of FIRE does not affect responsiveness to IL3. By 2 weeks post-differentiation, a lawn of ESC-derived macrophages are produced from the *Csf1r*$^{+/+}$ EB, whereas the number of macrophages from *Csf1r*$^{\Delta FIRE/\Delta FIRE}$ clones was ≈200-fold lower (Fig. 1f). Overall, these in vitro results indicate that FIRE is required for expression of *Csf1r* and/or macrophage differentiation in EB.

**Generation of *Csf1r*$^{\Delta FIRE/\Delta FIRE}$ mice.** To produce *Csf1r*$^{\Delta FIRE/\Delta FIRE}$ mice, vectors encoding the FIRE gRNAs US1 and DS2 and Cas9 were microinjected into C57BL/6J/CBA F1 mouse oocytes. One male founder with the expected 418 bp deletion was crossed with *Csf1r*$^{+/+}$ C57BL/6J females and the progeny were interbred. The frequencies of *Csf1r*$^{+/+}$, *Csf1r*$^{+/\Delta FIRE}$ and *Csf1r*$^{\Delta FIRE/\Delta FIRE}$ mice at weaning are 26%, 55%, and 19%, respectively ($n = 501$). Unlike *Csf1r*$^{-/-}$ mice or the *Csf1*$^{op/op}$ mouse[4] there is no postnatal growth retardation, failure of tooth eruption or evidence of osteopetrosis in *Csf1r*$^{\Delta FIRE/\Delta FIRE}$ mice (Fig. 2a–c). The proximal growth plates of control and *Csf1r*$^{\Delta FIRE/\Delta FIRE}$ mice are indistinguishable and normal numbers of osteoclasts are present (Fig. 2d, e). Brain abnormalities

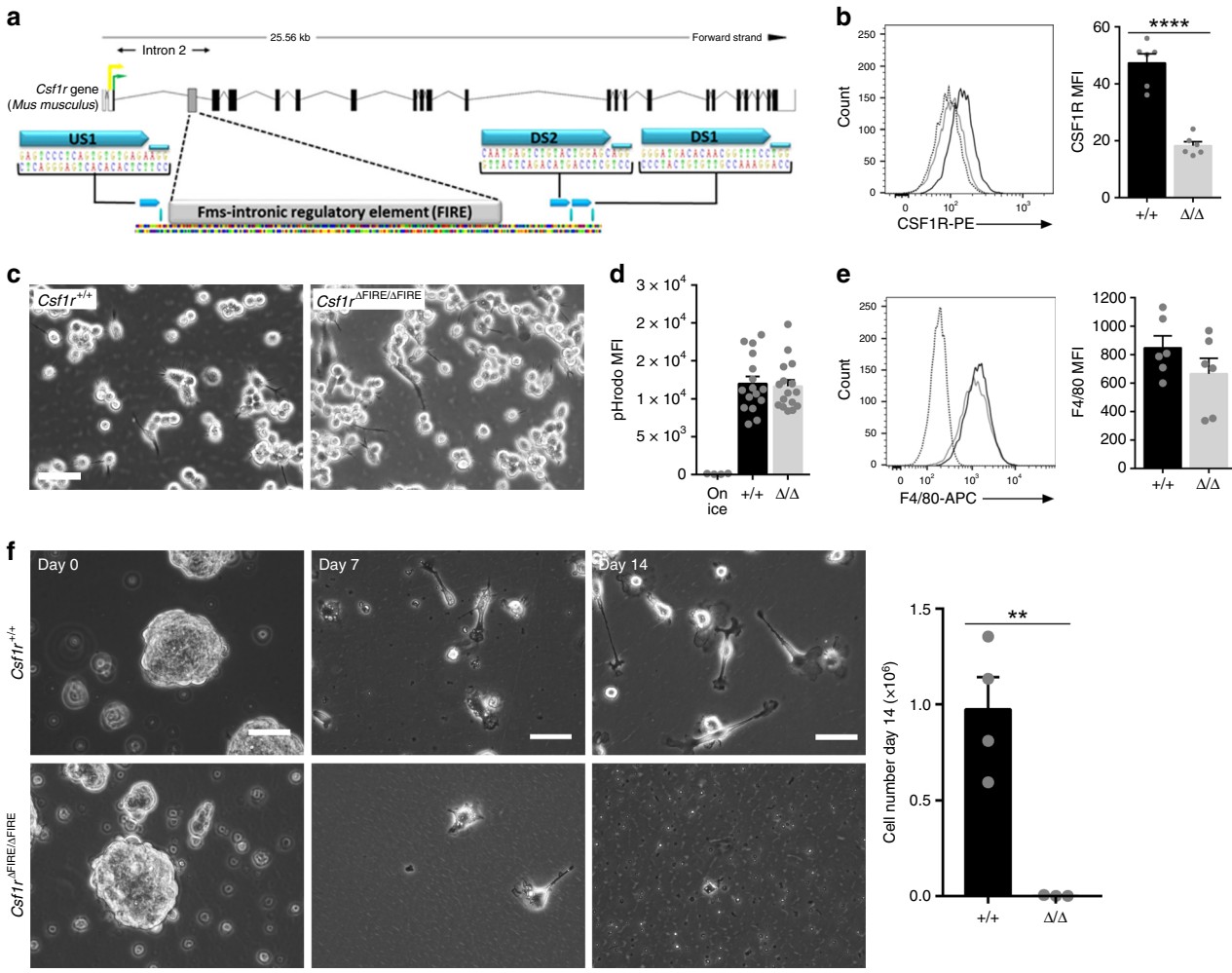

**Fig. 1** FIRE deletion reduced CSF1R expression and prevented macrophage differentiation. **a** Schematic of *Csf1r* gene and location of gRNAs (US1, DS1, DS2), targeting FIRE. Protospacer adjacent motifs (PAM) are located to the right of each gRNA. Black boxes = coding exons, white boxes = untranslated regions, lines connecting exons = intronic sequences, yellow and green arrows = trophoblast/osteoclast promoter and macrophage transcription start sites, respectively. **b** Flow cytometry analysis of total CSF1R in fixed and permeabilized RAW 264.7 cell clones. Histograms are representative of three clones per genotype, from two repeat experiments in duplicate. MFI median fluorescence intensity, black line = +/+, gray line = Δ/Δ, dotted line = isotype control. $P \leq 0.0001$ (****). **c** CRISPR-treated *Csf1r*^+/+ and *Csf1r*^ΔFIRE/ΔFIRE RAW 264.7 cells cultured in vitro. Scale bar = 50 μm. **d** A phagocytosis assay was performed with RAW 264.7 cells and pHrodo® Red *E. coli* BioParticles® followed by flow cytometry analysis. $n = 4$ clones per genotype from two repeat experiments. MFI median fluorescence intensity. **e** Flow cytometry analysis of surface F4/80 in RAW 264.7 cell clones. Histograms are representative of three clones per genotype from two repeat experiments in duplicate. **f** *Csf1r*^+/+ and *Csf1r*^ΔFIRE/ΔFIRE E14 ESC clones were differentiated into macrophages with mIL3 and rhCSF1. Scale bars = 200 μm (Day 0, embryoid bodies) and 50 μm (Days 7 and 14). $n = 3$ clones per genotype, $P = 0.005$ (**). All source data are provided within a Source Data excel file. Graphs show mean + SEM and $P$ values were determined by two-tailed *t*-tests

in *Csf1r*^−/− mice include reduced brain size, severely enlarged ventricles, hollow olfactory bulbs, deficient myelination and failure of commissure closing in the corpus callosum[19]. The *Csf1r*^ΔFIRE/ΔFIRE brains are not macroscopically distinguishable from controls and Luxol fast blue staining reveals no differences in myelination (Fig. 2f, g).

*Csf1*^op/op, *Csf1r*^−/− mice and adult mice treated with anti-CSF1R antibody lack lamina propria and Peyer's patch macrophages in the small intestine. These mice also show associated changes in villus architecture, increased goblet cell numbers detected with periodic acid-schiff (PAS) staining, reduced proliferation in the crypts and defects in differentiation of Paneth cells[6,20,21]. *Csf1r* mRNA is not expressed in intestinal epithelial cells and the impacts of anti-CSF1R treatment are attributable to the loss of macrophages surrounding the crypts[21]. There are no detectable alterations in intestinal epithelia that distinguish

between *Csf1r*^+/+ and *Csf1r*^ΔFIRE/ΔFIRE mice (Fig. 2h and Supplementary Fig. 2).

The development of normal sexual maturity is compromised in both male and female *Csf1*^op/op mice[22,23] and in *Csf1r*^−/− rats[5]. By contrast, the *Csf1r*^ΔFIRE/ΔFIRE mice reach sexual maturity and are fertile.

**The impact of FIRE deletion in blood and BM.** Two subsets of blood monocytes can be distinguished based upon the expression of LY6C. The differentiation of the LY6C^lo subset depends upon CSF1R signaling[6,24]. *Csf1r*^ΔFIRE/ΔFIRE mice have normal cell populations in peripheral blood (Fig. 3a) and F4/80, LY6C, and CD11b expression is unchanged compared to littermates (Fig. 3b–d). However, the expression of CSF1R detected with anti-CD115 antibody is abolished (Fig. 3c). Furthermore, the binding of labeled CSF1 (pCSF1-Fc^AF647)[11] is undetectable in the mutant mice, whereas in littermates it provides a sensitive marker

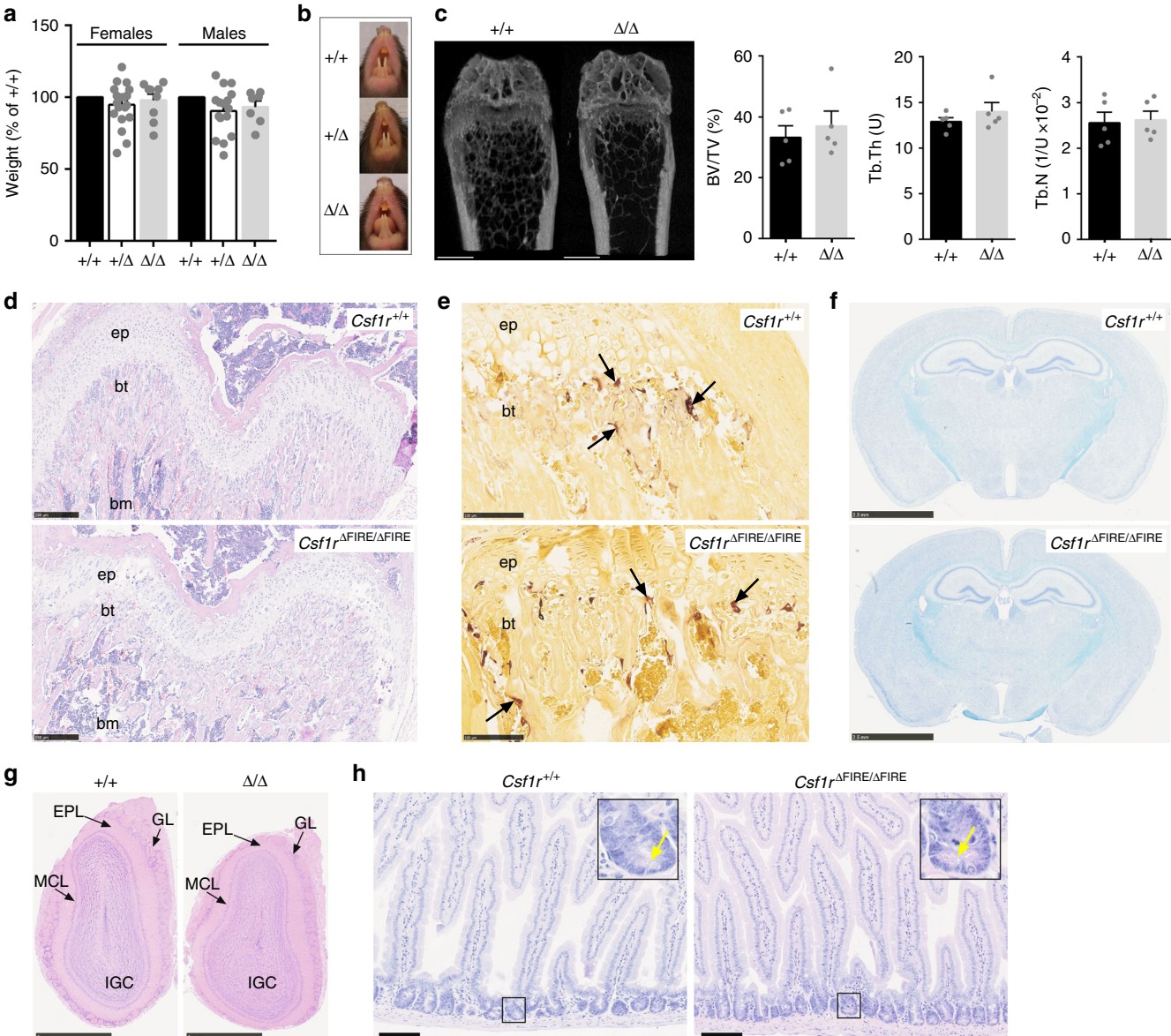

**Fig. 2** FIRE-deficient mice displayed none of the gross phenotypes found in *Csf1r*$^{-/-}$-mice. **a** Mice were weighed between 2 and 11 weeks of age. $n = 7$, 20, 9 (females) and 5, 16, 7 (males) for $+/+$, $+/\Delta$, and $\Delta/\Delta$, respectively. **b** Representative images of tooth eruption in adult mice. **c** Femurs from 4-week-old male and female mice were scanned by Micro-CT. Scale bars $= 1$ mm. $n = 5$ mice per genotype. BV/TV bone volume over total volume (bone density), Tb. Th trabecular thickness, Tb.N trabecular number. **d** Hematoxylin and eosin (H&E) staining of sections of bone. Images are representative of six mice per genotype; ep epiphyseal plate, bt bony trabeculae, bm bone marrow. Scale bar $= 250$ μm. **e** Femurs from the same cohort of mice above were stained for tartrate-resistant acid phosphatase (TRAP). Arrows point to TRAP$^+$ osteoclasts. Scale bar $= 100$ μm. **f** Luxol Fast Blue staining of paraffin-embedded formalin-fixed adult brains. Images are representative of three mice per genotype. Scale bar $= 2.5$ mm. **g** H&E staining of olfactory bulbs from adult mice. Images are representative of six mice per genotype. EPL external plexiform layer, GL glomerular layer, IGC internal granular cell layer of the olfactory bulbs, MCL mitral cell layer. Scale bar $= 1$ mm. **h** H&E staining of sections of duodenum from adult mice. Yellow arrows point to Paneth cells. Images are representative of six mice per genotype. Scale bar $= 100$ μm. All source data are provided within a Source Data excel file. Graphs show mean $+$ SEM

for F4/80$^+$ blood monocytes (Fig. 3d). Notably, the level of both CSF1R and CSF1 binding is reduced by around 50% in the heterozygotes (Fig. 3c, d), which is consistent with a lack of dosage compensation at the mRNA level also reported in *Csf1r*-deficient mice and rats[4,5].

Unlike *Csf1r* and *Csf1* mutant mice[4], *Csf1r*$^{\Delta FIRE/\Delta FIRE}$ mice are not osteoclast-deficient and osteopetrotic. Total BM cellularity remains unchanged by the FIRE mutation (Supplementary Fig. 3a). Aside from progenitors, BM contains multiple specialized macrophage populations that regulate bone homeostasis and hematopoietic differentiation (reviewed in ref. [25]). BM cell populations were

analyzed by flow cytometry as described in Supplementary Fig. 3b. Two populations of F4/80$^+$ cells are detected in control mice that differed in the level of CSF1R. The proportion of F4/80$^+$ cells is unchanged in *Csf1r*$^{\Delta FIRE/\Delta FIRE}$ mice, but CSF1R expression and CSF1 binding is abolished in both F4/80$^+$ populations (Supplementary Fig. 3c and d). As in the blood, the level of CSF1R is reduced around 50% in heterozygotes. BM contains a population of monocyte–dendritic cell progenitors (MDP) that express KIT, FLT3, and CSF1R, and committed proliferative LY6C$^+$ progenitors derived from the MDP that lack both KIT and FLT3[26]. Although CSF1R expression is lost in *Csf1r*$^{\Delta FIRE/\Delta FIRE}$ BM, there is no

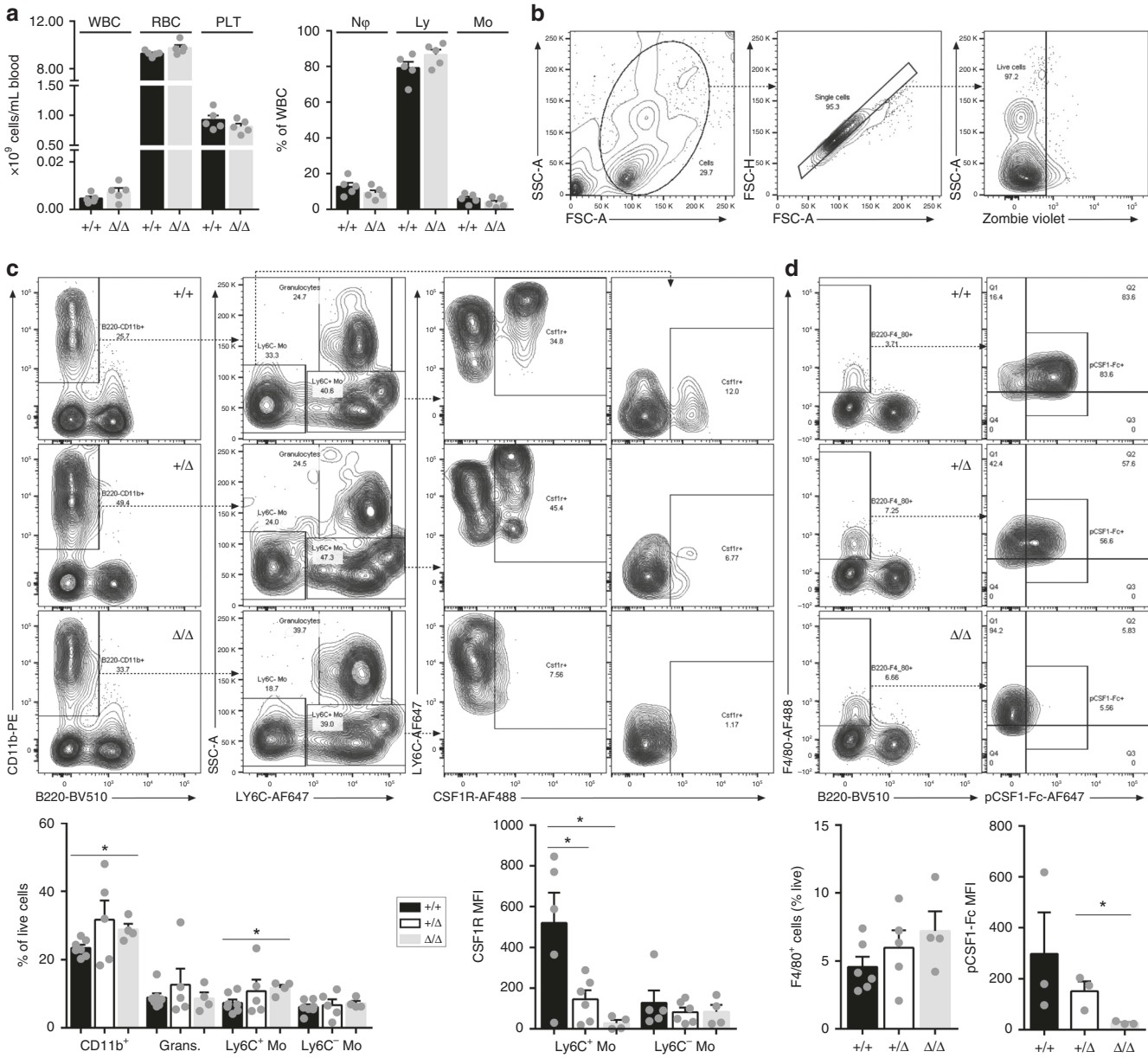

**Fig. 3** FIRE deletion reduced functional CSF1R expression in blood. **a** Cell populations in whole blood of 11–13-week-old $Csf1r^{\Delta FIRE/\Delta FIRE}$ mice ($\Delta/\Delta$) and controls ($+/+$) were quantified using an automated hematology analyzer. $n = 5$ mice per genotype. WBC white blood cells, RBC red blood cells, PLT platelets, Nφ neutrophils, Ly lymphocytes, Mo monocytes. **b** Whole EDTA-blood from 11 to 13-week-old mice was analyzed by flow cytometry. Plots show the representative gating strategy to identify live single cells for subsequent analysis. **c** Myeloid cells were identified as CD11b+B220− (Panel 1) then separated by LY6C and SSC to identify granulocytes (LY6C+SSC[hi], panel 2). The LY6C+ and LY6C[lo] monocyte populations were then analyzed for CSF1R expression (Panels 3 and 4, respectively). $n = 7+/+$, $5+/\Delta$, and $4 \Delta/\Delta$ from four experiments. For % of live cells $P = 0.012$ (*CD11b+) and 0.019 (*LY6C+ Mo). For median fluorescence intensity (MFI) of CSF1R $P = 0.027$ (*+/+ vs. +/$\Delta$), and 0.022 (*+/+ vs. $\Delta/\Delta$). Grans. granulocytes, Mo monocyte. **d** F4/80+B220− myeloid cells were analyzed for binding of CSF1 (pCSF1-Fc[AF647])[11]. $n = 3$ mice per genotype from four experiments. $P = 0.033$ (*). All source data are provided within a Source Data excel file. Graphs show mean + SEM and $P$ values were determined by two-tailed $t$-tests

reduction in the number of cells expressing LY6C, KIT, or FLT3 (Supplementary Fig. 3e). In view of the apparent loss of CSF1R protein, we examined whether BM from mutant mice is responsive to CSF1. BM-derived macrophages (BMDM) are generated from $Csf1r^{\Delta FIRE/\Delta FIRE}$ BM, but with greatly reduced yield compared to controls (Supplementary Fig. 3f). The macrophages derived from $Csf1r^{\Delta FIRE/\Delta FIRE}$ BM still express F4/80 and CD11b but have lower levels of surface CSF1R detected with pCSF1-Fc[AF647] relative to controls (Supplementary Fig. 3g). Hence, the production/survival of these few BMDM relies on some mechanism that compensates for the absence of FIRE.

**Macrophage populations in $Csf1r^{\Delta FIRE/\Delta FIRE}$ mice**. We performed flow cytometry and/or immunohistochemistry analyses on major organs as shown in Fig. 4, Supplementary Figs. 4 and 5. Liver macrophages (Kupffer cells, KC) are CSF1R-dependent[4,6], CSF1-responsive[27], and the main site of clearance of labeled CSF1[11]. The number, location, or morphology of KC and the liver-to-body weight ratio of $Csf1r^{\Delta FIRE/\Delta FIRE}$ mice are similar to littermate controls (Supplementary Fig. 4a, b). Flow cytometry analysis confirmed the unchanged numbers of KC and dendritic cell subsets (cDC2 as defined previously[11]) (Fig. 4a) and retention of KC-specific markers such as TIM4 (Supplementary Fig. 4c).

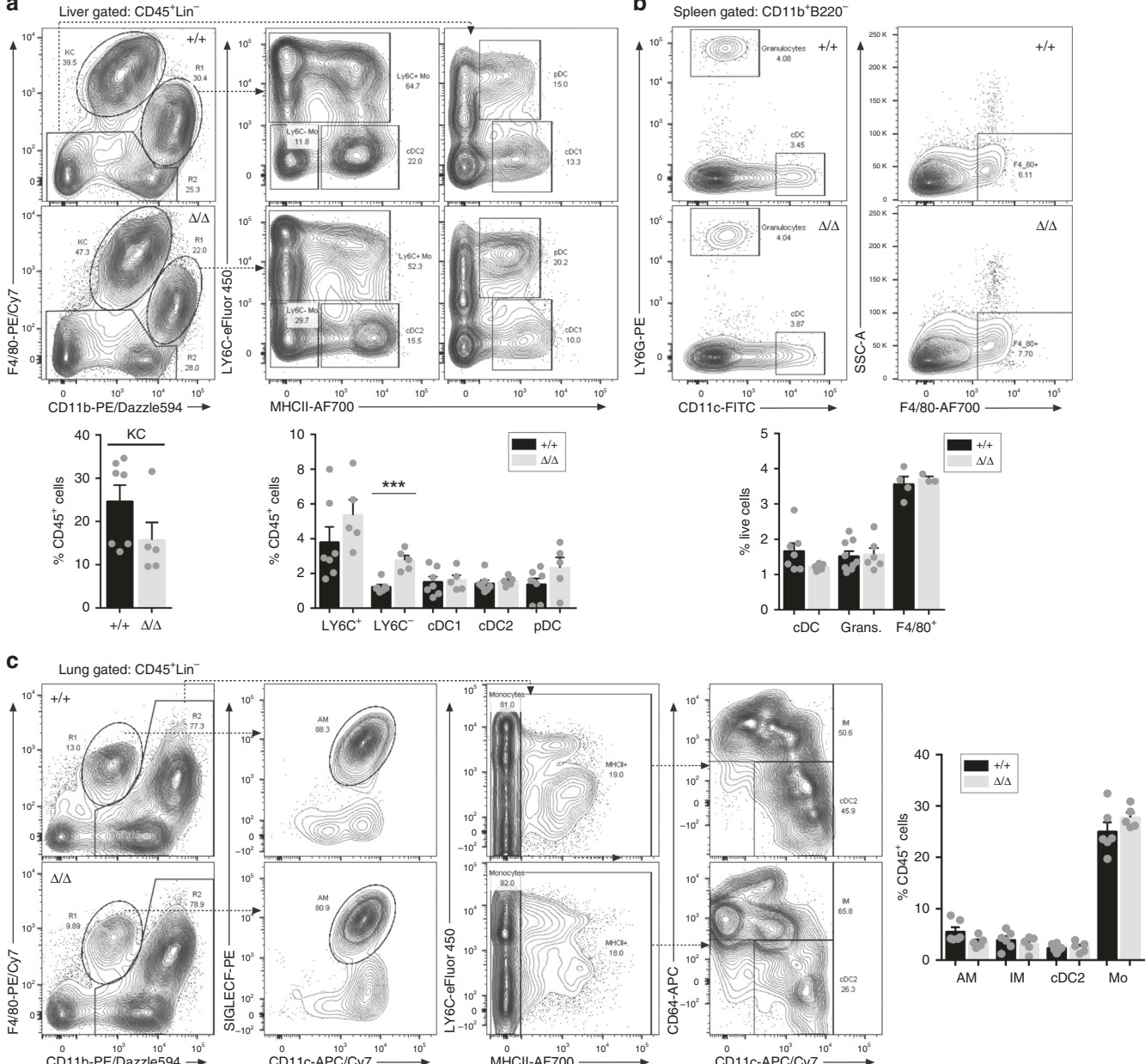

**Fig. 4** Macrophage populations in liver, spleen, and lungs are unaffected in FIRE-deficient mice. **a** Leukocyte populations were analyzed from enzymatically digested livers from mice aged between 8 and 10 weeks. Flow cytometry profiles show representative $Csf1r^{+/+}$ ($+/+$) and $Csf1r^{\Delta FIRE/\Delta FIRE}$ ($\Delta/\Delta$) mice using the gating strategy described by Hawley et al.[11]. Kupffer cells (KC) were identified as CD45$^+$, lineage$^-$ (Lin-; CD3$^-$CD19$^-$LY6G$^-$) and F4/80$^{hi}$CD11b$^{lo}$ (Panel 1). F4/80$^{lo}$/CD11b$^{hi}$ and F4/80$^-$/CD11b$^-$ populations were further separated on the basis of LY6C and MHCII expression (Panels 2 and 3). Mo monocyte, cDC conventional dendritic cell, pDC plasmacytoid dendritic cell. $n = 7+/+$ and 5 $\Delta/\Delta$ from 3 repeat experiments. $P = 0.0002$ (***). **b** Leukocyte populations were analyzed from enzymatically digested spleens from mice aged between 8 and 10 weeks. The flow cytometry profiles show representative $Csf1r^{+/+}$ ($+/+$) and $Csf1r^{\Delta FIRE/\Delta FIRE}$ ($\Delta/\Delta$) mice. Conventional dendritic cells (cDC) were identified as CD11b$^+$B220$^-$CD11c$^+$LY6G$^-$ and granulocytes as CD11b$^+$B220$^-$CD11c$^-$LY6G$^+$ (Panel 1). Red pulp macrophages were identified as CD11b$^+$B220$^-$F4/80$^+$SSC$^{lo}$ (Panel 2). $n = 3$–9 mice per genotype from 4 repeat experiments. **c** Single cell suspensions of digested lungs were analyzed by flow cytometry as described above for the liver using the same cohort of mice. Alveolar macrophages (AM) were defined as F480$^+$CD11b$^{lo}$ (Panel 1) and CD11c$^+$SIGLECF$^+$ (Panel 2). The CD11b$^+$F4/80$^{-/+}$ cells were further analyzed to define the monocytes (Mo, MHCII$^-$LY6C$^{-/+}$, Panel 3), interstitial macrophages (IM, MHCII$^+$CD11c$^{-/+}$CD64$^+$) and conventional dendritic cells subset 2 (cDC2, MHCII$^+$CD11c$^+$CD64$^-$) in Panel 4. All source data are provided within a Source Data excel file. Graphs show mean + SEM and $P$ values were determined by two-tailed $t$-tests

However, there is a significant increase in LY6C$^{lo}$ monocyte-like cells (CD11b$^+$F4/80$^{lo}$). The full gating strategy for liver flow cytometry data is shown in Supplementary Fig. 4d.

The marginal zone is completely absent in the spleen of $Csf1^{op/op}$ mice[28] and in $Csf1r^{-/-}$ rats[5]. In $Csf1r^{\Delta FIRE/\Delta FIRE}$ mice the overall architecture and relative area of the marginal zone is indistinguishable from the controls (Supplementary Fig. 4e and f). As in the liver, there is no change in the spleen-to-body weight ratio or the relative abundance of myeloid populations defined by staining for F4/80$^+$ or other markers (Supplementary Fig. 4g, h and Fig. 4b). The full gating strategy for spleen flow cytometry data is shown in Supplementary Fig. 4i.

 

Lung macrophage populations are primarily controlled by CSF2 (also known as GM-CSF, encoded by *Csf2*), rather than CSF1[29], but lung macrophages do express *Csf1r* and bind CSF1[11]. The *Csf1r*-EGFP macrophage-reporter transgene is highly expressed in both alveolar (AM) and interstitial macrophage populations in the lung[10], which have distinct developmental origins and renewal mechanisms during adulthood[30]. Furthermore, expression of *Csf1r*-EGFP in AM is dependent upon the inclusion of FIRE in the transgene[10]. Macrophages of the lung are partly depleted in *Csf1*[op/op] mice[31] and double mutant *Csf1*[op/op] and *Csf2*[−/−] mice develop severe alveolar proteinosis[29]. AM can be identified based upon expression of SIGLECF[30]. Interstitial myeloid populations have been classified as macrophage and conventional dendritic cell subpopulations, based upon expression of CD64, CD11b, CD11c, and MHCII[11]. To assess the impact of the FIRE deletion on these populations, lungs were digested and analyzed by flow cytometry as previously described[11]. The full gating strategy is shown in Supplementary Fig. 5a. We also measured *Csf1r* mRNA in the lung of wild-type and *Csf1r*[ΔFIRE/ΔFIRE] mice (Supplementary Fig. 5b). FIRE deletion has no effect on AM or any of the interstitial myeloid populations of the lung (Fig. 4c). A marginal reduction in *Csf1r* mRNA is likely attributable to the loss of expression in monocytes in both blood and BM (Fig. 3c, d and Supplementary Fig. 3c, d). F4/80 is barely detectable in either of the major lung macrophage populations and is primarily associated with monocytes[30]. Accordingly, the levels of total *Adgre1* (encoding F4/80) and *Itgam* (CD11b) mRNA in the lung are unaffected by the deletion of FIRE. We conclude that FIRE is not required for expression of *Csf1r* mRNA in lung macrophages nor for their development or monocyte trafficking in the lung.

The large macrophage population of the intestine is constantly renewed from recruited blood monocytes[32]. The Peyer's patch and surrounding mucosa contain several specialized mononuclear phagocyte populations, distinguished by mRNA or protein expression of the surface markers F4/80 (ADGRE1), CD4, SIGLEC1, SIRPA, MERTK, TIM4, CX3CR1, and ITGAX (CD11c)[33], including a recently described long-lived population[34,35]. All of these populations depend upon continuous CSF1R signaling[6,21]. Monocytes enter the lamina propria constantly and respond to the local environment with progressive changes in surface phenotype and gene expression in a so-called monocyte waterfall[32]. Supplementary Fig. 6 shows the comparison of these populations in the *Csf1r*[+/+] and *Csf1r*[ΔFIRE/ΔFIRE] mice. There is no detectable difference in distribution of monocyte-like, DC-like, or macrophage populations in either small or large intestine.

Macrophages are so abundant in the intestine that the level of macrophage-specific transcripts can be reproducibly detected by analysis of total mRNA. To complement the analysis of isolated cells, we isolated mRNA from Peyer's patches and surrounding mucosa in the ileum of *Csf1r*[+/+] and *Csf1r*[ΔFIRE/ΔFIRE] mice and profiled gene expression using microarrays. There is no significant difference in *Csf1r* expression between *Csf1r*[+/+] and *Csf1r*[ΔFIRE/ΔFIRE] mice nor in the level of mRNA encoding any of the markers of intestinal macrophage subpopulations mentioned above, including *Itgam* (CD11b) and *Ly6c1*, which are rapidly down-regulated following extravasation into the intestine[32] (Supplementary Data 1). These findings demonstrate that FIRE is not required by intestinal macrophages for expression of *Csf1r* and that the loss of CSF1R in monocytes has no impact on intestinal macrophage homeostasis. In addition, the expression of transcripts that mark cell types controlled indirectly by signals from CSF1R-dependent macrophages, including markers of Paneth cells (*Lyz1*), microfold cells (*Spib, Ccl9, Tnfrsf11a*) or goblet cells (*Muc2*)[21], are unaffected (Supplementary Data 1).

**Loss of macrophage populations in *Csf1r*[ΔFIRE/ΔFIRE] mice.** The resident macrophages of the peritoneal cavity may be subdivided into two populations, based upon relative size and the level of TIM4 and F4/80 antigen they express[36]. Both populations of peritoneal macrophages (PM) are absent from *Csf1*[op/op] and *Csf1r*[−/−] mice[4,28] and from mice treated with anti-CSF1R antibody[6]. The major TIM4[+]F4/80[hi]CD11b[hi] large PM population is almost entirely lost in *Csf1r*[ΔFIRE/ΔFIRE] mice (Fig. 5a and Supplementary Fig. 6a). The minor residual PM population consists of the small PM that are TIM4[−]F4/80[lo][36]. These cells were shown previously to bind labeled CSF1 preferentially compared to large PM[11]. Both anti-CSF1R (CD115) and CSF1-Fc binding to these residual populations are absent in the *Csf1r*[ΔFIRE/ΔFIRE] mice (Fig. 5a). In *Csf1r*[ΔFIRE/+] mice the PM populations are retained but as in blood and BM, there is a reduction in the level of surface receptor detected with either labeled CSF1 or anti-CSF1R antibody, which is consistent with the lack of dosage compensation. In summary, in the peritoneum, the *Csf1r*[ΔFIRE/ΔFIRE] mutation phenocopies the *Csf1r*-deficient mice.

The abundant interstitial F4/80[hi] macrophage populations of the kidney have recognized roles in renal growth and development[37], homeostasis and immune surveillance[38]. F4/80[hi] kidney macrophages were almost absent from both *Csf1*[op/op] and *Csf1r*[−/−] mice[4]. To identify populations of renal mononuclear phagocytes we enzymatically digested kidneys and analyzed CD45[+] myeloid cells by flow cytometry. The gating strategy is shown in Supplementary Fig. 7b. *Csf1r*[ΔFIRE/ΔFIRE] mice have fewer total CD45[+] cells and the F4/80[hi]CD11b[lo] resident kidney macrophages (KM)[38] are undetectable (Fig. 5b). In both control and mutant, a subset of the CD11b[hi] cells are F4/80[lo]MHCII[−] monocytes but the majority of cells in the F4/80[lo]CD11b[hi] (R1) gate are also MHCII[+]/CD64[+]. Their surface phenotype is reminiscent of the monocyte-derived small PM discussed above. As in the peritoneum, their abundance is slightly increased by the FIRE mutation (Fig. 5b). Conventional dendritic cells (cDC) in the kidney are difficult to separate from macrophages based upon surface markers. A minor subpopulation of cDC (F4/80[−]CD11b[+]MHCII[+]) is detected in both control and *Csf1r*[ΔFIRE/ΔFIRE] kidneys (R2 gate) (Fig. 5b). Although exogenous CSF1 can promote mouse kidney morphogenesis in vitro and growth of the kidney in vivo[37], there is no change in the kidney-to-body weight ratio in *Csf1r*[ΔFIRE/ΔFIRE] mice (Supplementary Fig. 7c) and kidneys are histologically normal (Supplementary Fig. 7d).

Macrophages of the heart can also be separated into subpopulations attributed distinct roles in cardiac homeostasis and repair following injury[39–41]. Supplementary Fig. 7e shows the flow cytometry gating strategy for the heart and Fig. 5c summarizes the analysis of digested cardiac macrophage populations from control and *Csf1r*[ΔFIRE/ΔFIRE] mice. The total cardiac CD45[+] population is greatly reduced in the *Csf1r*[ΔFIRE/ΔFIRE] mice due to selective loss (70–80%) of both the F4/80[hi]CD64[+] macrophage (MHCII[hi]) and monocyte-like (MHCII[lo]) populations previously described by others[39–41]. Interestingly, a recent study proposed a link between resident macrophages of the kidney and the heart in the generation of an adaptive response to cardiac pressure overload[40]. However, as in the kidney, we detect no change in the heart-to-body weight ratio in *Csf1r*[ΔFIRE/ΔFIRE] mice (Supplementary Fig. 7f) and the histology is indistinguishable from controls (Supplementary Fig. 7g).

There are two major macrophage populations in the skin, the Langerhans cells of the epidermis and the dermal macrophages. Both are CSF1R-dependent[4,6] but Langerhans cells, like microglia, depend specifically on the alternative CSF1R ligand, IL34[42]. Langerhans cells are absent in epidermal sheets from tails of *Csf1r*[ΔFIRE/ΔFIRE] mice (Fig. 6a). The loss of this cell population

 

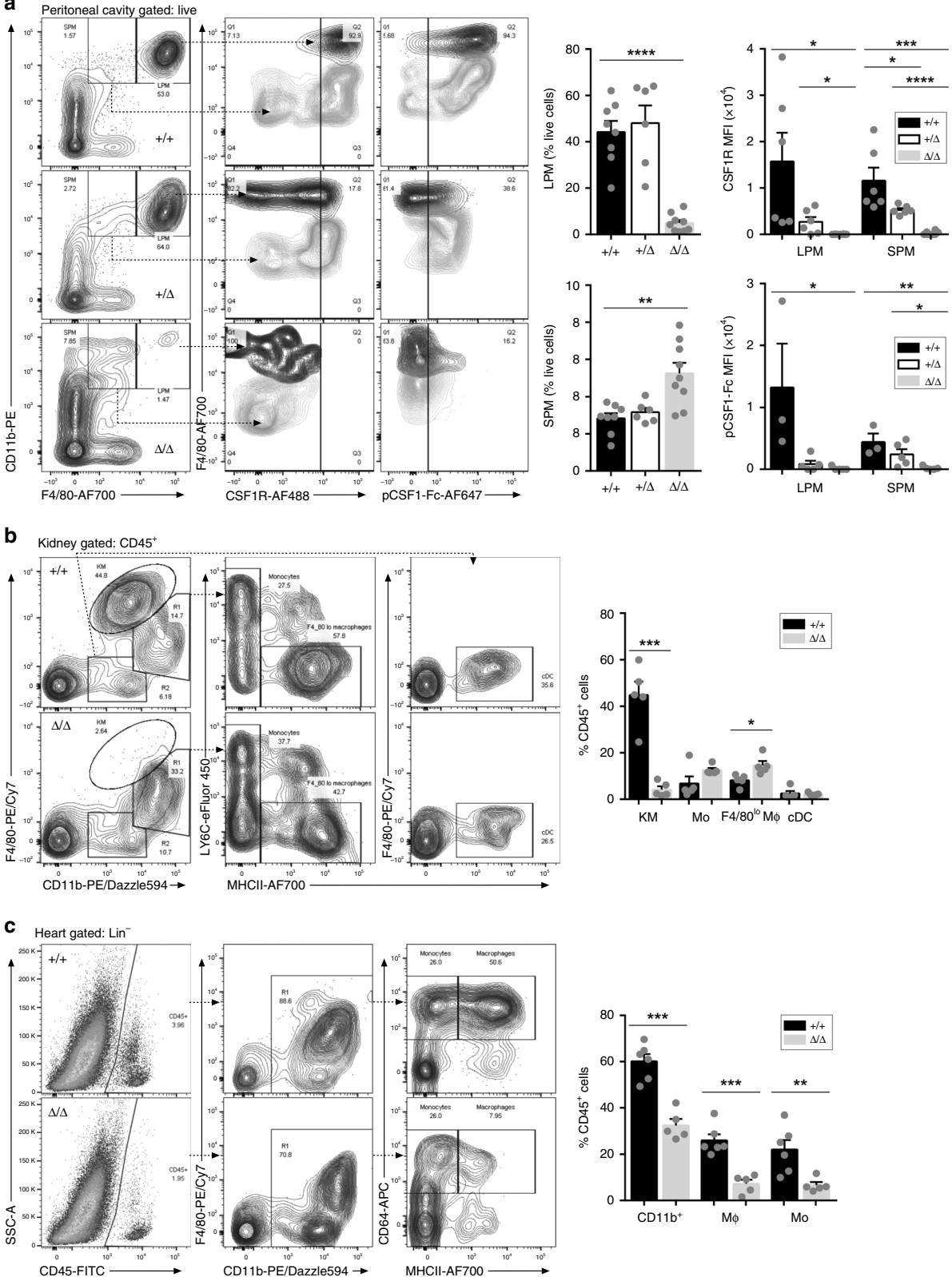

has no effect on the morphology of the epidermis or the dermis (Fig. 6b). Dermal macrophage populations accumulate melanosomes from neighboring melanocytes and, like macrophages of the gut, they turn over rapidly[43]. The stellate melanophagic macrophage populations of the dermis are F4/80+ and are not affected by the FIRE mutation (Fig. 6b).

**Loss of microglia in *Csf1r*[ΔFIRE/ΔFIRE] mice**. Microglia, the tissue-resident macrophage population of the brain, arise very early in mouse embryonic development and depend upon CSF1R signaling[44]. Microglia detected by localization of IBA1 or P2RY12 in littermate controls are absent in the brain of *Csf1r*[ΔFIRE/ΔFIRE] mice (Fig. 7a, b). Disaggregation of the whole

**Fig. 5** FIRE deletion results in loss of macrophages in the peritoneal cavity, kidney, and heart. **a** Flow cytometry profiles show peritoneal cells from representative $Csf1r^{+/+}$ and $Csf1r^{\Delta FIRE/\Delta FIRE}$ mice using the gating strategy described by Bain et al.[36]. Small peritoneal macrophages (SPM) were identified as F4/80[lo]CD11b[+] and large peritoneal macrophages (LPM) as F4/80[hi]CD11b[+] (Panel 1). These populations were analyzed for CSF1R expression (Panel 2) and binding of pCSF1-Fc (Panel 3). $n = 8 +/+$, $6+/\Delta$ and $9 \Delta/\Delta$ mice aged between 10 and 15 weeks, from five experiments. $P = 0.012–0.048$ (*), $0.002–0.006$ (**), $0.001$ (***) and $<0.0001$ (****). **b** Leukocyte populations were isolated from enzymatically digested kidneys and gated for CD45 expression. The flow cytometry profiles show analysis of representative $Csf1r^{+/+}$ and $Csf1r^{\Delta FIRE/\Delta FIRE}$ mice. Kidney macrophages (KM) were identified as F4/80[hi]CD11b[lo] (Panel 1), pooled LY6C[−/+] monocytes (Mo) as F4/80[lo]MHCII[−] (Panel 2), putative monocyte-derived macrophages as F4/80[lo]MHCII[+] (Panel 2), and conventional dendritic cells (cDC) as F4/80[−]CD11b[lo]MHCII[+] (Panel 3). $n = 5$ mice per genotype aged 8–10 weeks, from three experiments. $P = 0.015$ (*) and $0.0001$ (***). **c** Single cell suspensions of enzymatically digested hearts from the same cohort were analyzed by flow cytometry. Cells were gated lineage[−] (Lin[−] = CD3/CD19/LY6G). $n = 6+/+$ and $5 \Delta/\Delta$ from three experiments. MΦ macrophage, Mo monocyte. $P = 0.009$ (**), $0.0003$ (*** MΦ), and $0.0001$ (*** CD11b[+]). All source data are provided within a Source Data excel file. Graphs show mean + SEM and $P$ values were determined by two-tailed $t$-tests

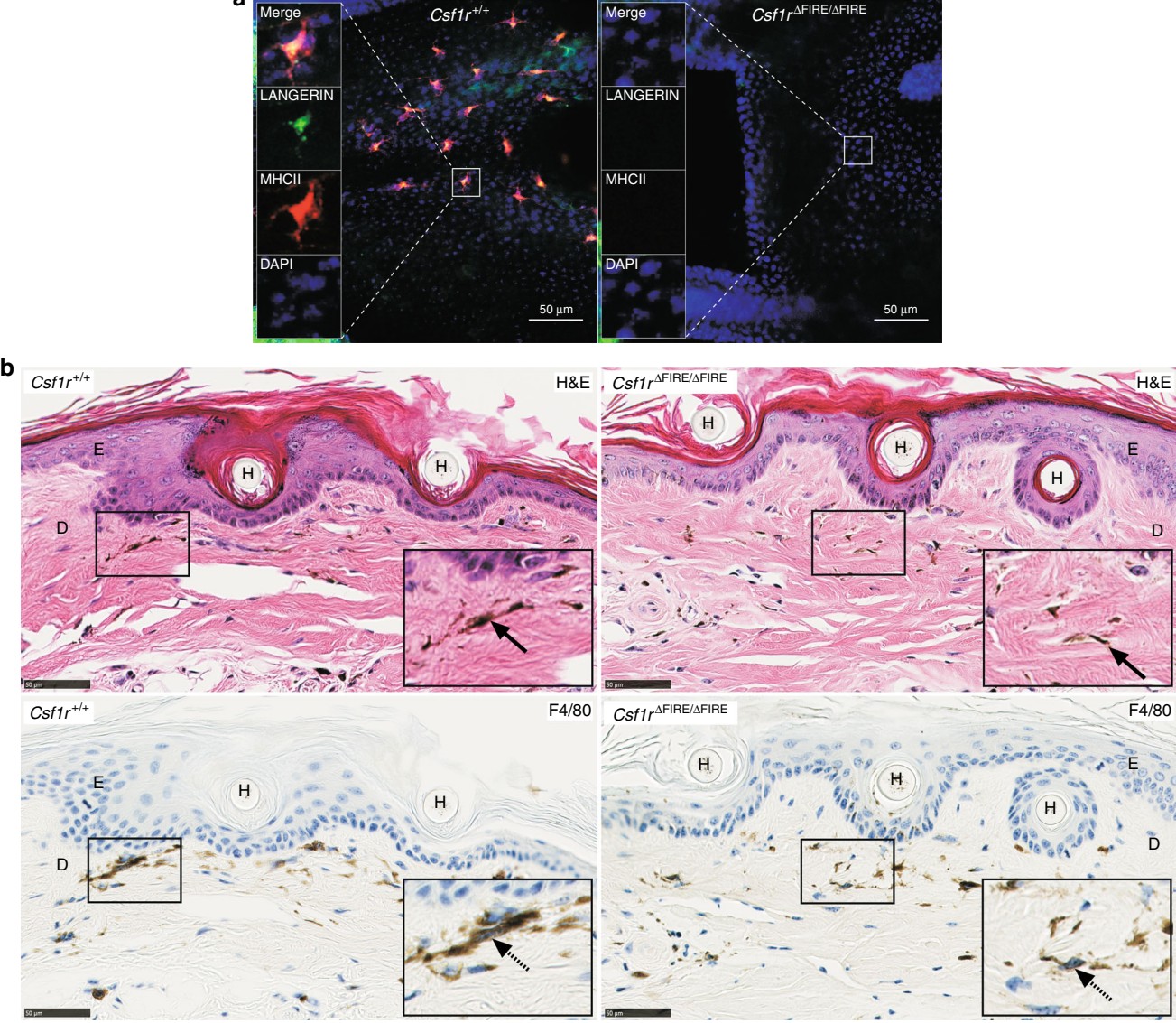

**Fig. 6** The deletion of FIRE results in loss of Langerhans cells. **a** Whole-mount epidermal tail sheets were stained with antibodies against LANGERIN (green) and MHCII (red). Nuclei were stained with DAPI (blue). Images are representative of four mice per genotype, aged between 8 and 10 weeks. Scale bars = 50 μm. **b** Decalcified tails from the same cohort of mice above were stained with hematoxylin and eosin (H&E, upper panel) and an antibody against F4/80 (lower panel). E epidermis, D dermis, H hair follicle. Scale bars = 50 μm

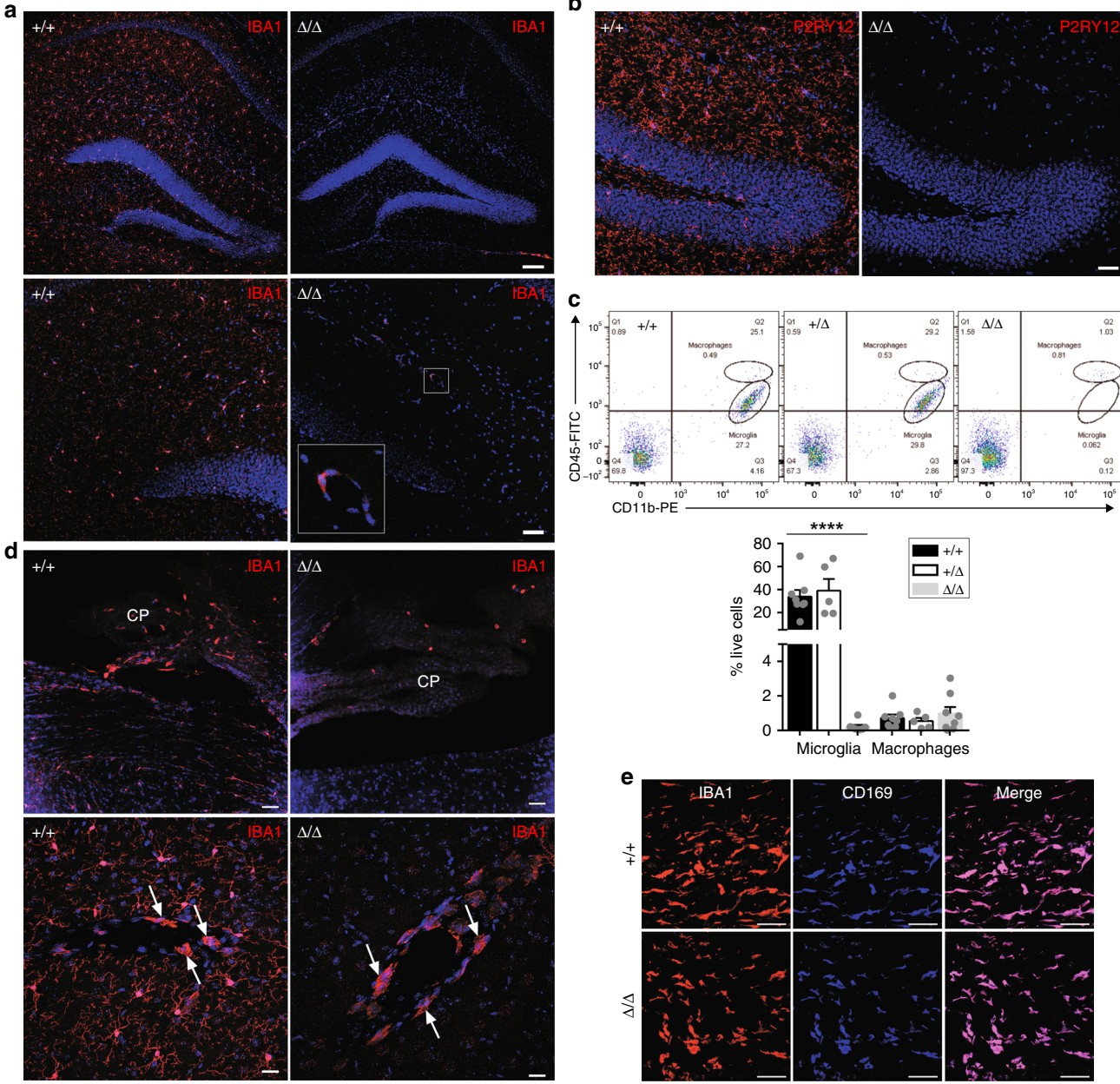

**Fig. 7** FIRE deletion results in selective loss of microglia. **a** Cryosections of adult brains were stained with an antibody against IBA1. Boxed area shows an IBA1+ perivascular macrophage in the *Csf1r*^ΔFIRE/ΔFIRE mouse brain. Scale bars = 100 μm, 50 μm in the upper and lower panels, respectively. Images are representative of four mice per genotype. **b** Cryosections of adult brains were stained with an antibody against P2RY12. Scale bar = 40 μm. **c** Single cell suspensions of myelin-depleted brains from males and females aged 10 weeks to 9 months were analyzed by flow cytometry for CD45 and CD11b expression as per[45]. Microglia = CD45^lowCD11b+. Macrophages = CD45+CD11b+. *n* = 8+/+, 5+/Δ, 8 Δ/Δ from five experiments. *P* < 0.0001 (****) Δ/Δ compared to +/+. **d** Cryosections of adult brains stained with an antibody against IBA1 using the same cohort of mice in **a** and **b**. Upper panel shows the choroid plexus (CP) and white arrows point to perivascular macrophages in the lower panel. Images are representative of three mice. Scale bars = 40 and 20 μm in the upper and lower panels, respectively. **e** Dissected meningeal dura mater was stained with antibodies against IBA1 and CD169. Images are representative of two mice per genotype. Scale bars = 50 μm. All source data are provided within a Source Data excel file. Graphs show mean + SEM and *P* values were determined by two-tailed *t*-tests

brain and analysis by flow cytometry[45] confirms the loss of CD45^low/CD11b+ microglia which persists up to 9 months of age (Fig. 7c). The full flow cytometry gating strategy is shown in Supplementary Fig. 8a. Heterozygous mutation (haploinsufficiency) of *Csf1r* has been proposed as a model for the human neurodegenerative disease ALSP (adult onset leukoencephalopathy with axonal spheroids and pigmented glia)[46]. However, there is no detectable effect of the heterozygous mutation in mice (*Csf1r*^ΔFIRE/+) on microglial numbers. The brain also contains

populations of CD45^hi classical macrophages associated with the vasculature, meninges, and choroid plexus with a distinct transcriptional profile[47,48]. CD45^hi cells are retained in *Csf1r*^ΔFIRE/ΔFIRE brain digests (Fig. 7c) This was confirmed by immunolocalization of CD169, shown recently to be a marker alongside CD206 and LYVE1 for a perivascular macrophage population in most tissues[49] (Fig. 7d, e). The brain-to-body weight ratios in adult male and female *Csf1r*^ΔFIRE/ΔFIRE mice remains unchanged (Supplementary Fig. 8b).

Microglia are so abundant in the brain that their gene expression profile is readily detected in total mRNA. The complete loss of microglia in $Csf1r^{-/-}$ rat brains was associated with commensurate loss of known microglia-associated transcripts in four brain regions[5]. To confirm the loss of microglia we compared the gene expression profiles of $Csf1r^{\Delta FIRE/\Delta FIRE}$ and $Csf1r^{+/+}$ hippocampi. The results, including ANOVA analysis of significantly regulated transcripts, are contained within Supplementary Data 2. Overall, 85 genes are significantly downregulated in the $Csf1r^{\Delta FIRE/\Delta FIRE}$ mouse hippocampus and 77 genes from this group (including $Csf1r$) have been previously shown to be expressed in human and mouse microglia[45,50,51]. Of the top 35 transcripts found by Elmore and colleagues to be down-regulated in the brains of adult mice treated with a CSF1R kinase inhibitor[52], 26 are also significantly reduced in the hippocampus of $Csf1r^{\Delta FIRE/\Delta FIRE}$ mice. The gene set affected by the $Csf1r^{\Delta FIRE/\Delta FIRE}$ mutation includes a small set of novel genes that may be implicated in microglial biology. Amongst these, the immune checkpoint protein, VISTA ($Vsir$), was recently localized in both human and mouse and was attributed functions in CNS pathologies[53]. In addition, HVCN1 was previously shown to regulate the production of microglial reactive oxygen species[54]. Consistent with the retention of the, CD45$^{hi}$CD11b$^+$CD169$^+$ brain macrophage population in the $Csf1r^{\Delta FIRE/\Delta FIRE}$ mice (Fig. 7b–e), $Ptprc$ (encoding CD45 antigen) and myeloid markers, such as $Mrc1(Cd206)$, $Siglec1$, $Siglecf$, $Cd163$, $Csf2ra/b/b2$, $Fcgr2b$, $Tnfrsf11a$ (RANK) that have all been associated with these populations[47,48] are not significantly affected by the deletion of FIRE (Supplementary Data 2). The complete loss of detectable $Csf1r$ expression implies that the perivascular and brain-associated macrophages, like blood monocytes and the macrophages of the BM, have lost $Csf1r$ expression but are nevertheless produced normally.

Microglia have been implicated in oligodendrocyte maturation and synaptic plasticity[55]. The loss or induced depletion of microglia in other models leads to alterations in synaptic pruning, hippocampal neurogenesis, myelination, oligodendrocyte maturation, and astrocyte activation[56]. By contrast to phenotypes reported in $Csf1^{op/op}$ and $Csf1r^{-/-}$ mice[4], we observe no evidence of auditory or visual abnormalities in the $Csf1r^{\Delta FIRE/\Delta FIRE}$ mice. Sensory neuronal deficits have been associated with deficient hippocampal neurogenesis but in the $Csf1r^{\Delta FIRE/\Delta FIRE}$ mice there is no loss of expression of markers associated with neuronal progenitor cells (such as doublecortin ($Dcx$), and $Sox2$[57]) (Supplementary Data 2). Nandi et al.[58] claimed that $Csf1r$ is expressed in subsets of neuronal progenitors, and noted the relative loss of excitatory neurons, expressing $Cux1$ or $Ctip2$ (BCll1b) in the $Csf1r^{-/-}$ mouse brain. $CUX1$ mRNA was also reduced in the brain in a recently described human patient with a homozygous $CSF1R$ mutation[59]. Neither transcript is downregulated in the hippocampus of $Csf1r^{\Delta FIRE/\Delta FIRE}$ mice (Supplementary Data 2).

Zhang et al.[60] used expression profiles of isolated cells to identify a set of signatures of brain cell populations that can support deconvolution of whole brain expression data. As shown in Supplementary Fig. 8c, there is no significant effect of the $Csf1r^{\Delta FIRE/\Delta FIRE}$ mutation on any cell-type-specific signature (neuron, astrocyte, oligodendrocyte, endothelium) other than microglia. The lack of impact on astrocyte numbers or morphology was confirmed by immunolocalization of the specific markers GFAP and SOX9 in the hippocampus, as well as the striatum, olfactory bulbs, and cerebellum (Supplementary Fig. 8d). Whereas microglia have been implicated as major sources of trophic factors, mRNA expression of $Bdnf$, $Igf1$, $Tgfb1,2$, and $3$ is not affected by the absence of microglia in $Csf1r^{\Delta FIRE/\Delta FIRE}$ mice (Supplementary Data 2). Similarly,

macrophages and microglia selectively express very high levels of lysosomal-associated genes (e.g. $Lamp1$, $Ctsb$, $Atp6v0e$) and receptors for apoptotic cells (e.g. $Mertk$, $Axl$) compared to whole brain regions (e.g. see biogps.org) but the loss of microglia has no effect on the expression of any lysosome-associated transcripts in the hippocampus of $Csf1r^{\Delta FIRE/\Delta FIRE}$ mice. This conclusion is consistent with the proposal that astrocytes are potentially also engaged with apoptotic cell clearance[61] and might fill this role in the absence of microglia.

**Embryonic macrophage development in $Csf1r^{\Delta FIRE/\Delta FIRE}$ mice.** Many tissue resident macrophages are seeded during embryonic development from the yolk sac and fetal liver progenitors[24,30,41,44,62,63]. $Csf1r^{-/-}$ mice lack yolk sac-derived macrophages and microglia but develop fetal monocytes[44]. Treatment of a pregnant mouse with anti-CSF1R resulted in the loss of embryonic macrophages derived from the yolk sac, whereas circulating liver-derived monocytes were unaffected[63]. As shown in Fig. 8a the $Csf1r^{\Delta FIRE/\Delta FIRE}$ embryos develop normally but the overall density of IBA1$^+$ macrophages throughout the body is greatly reduced. In the critical window in which liver hematopoiesis is established and the blood–brain barrier is closed (embryonic day 12–13), IBA1$^+$ ameboid microglia are prevalent in controls but undetectable in $Csf1r^{\Delta FIRE/\Delta FIRE}$ embryos (Fig. 8b–d). Numerous macrophages expressing the CSF1R-ECFP and CSF1R-EGFP reporter genes line the ventricular and external surfaces of the developing brain from 9.5 dpc[10,64]. These cells are absent in $Csf1r^{\Delta FIRE/\Delta FIRE}$ embryos (Fig. 8b, c). Only occasional monocyte-like cells associated with blood vessels are seen in the brain parenchyma (Fig. 8b). By contrast, IBA1$^+$ monocytes/ macrophages are detected in the meninges and choroid plexus in both control and mutant embryos (Fig. 8c, d), consistent with our findings in the adult brain (Fig. 7). There is also a reduction in the IBA1$^+$ macrophages in the fetal liver at the same stages of development (Fig. 8e). We conclude that the $Csf1r^{\Delta FIRE/\Delta FIRE}$ mice recapitulate the phenotype of $Csf1r^{-/-}$ mice in early hematopoiesis, producing a selective deficiency of yolk sac-derived cells. The major macrophage populations that are retained in the adult $Csf1r^{\Delta FIRE/\Delta FIRE}$ mice (liver, lung, spleen, intestine, dermis, etc.) are established late in gestation and postnatally from monocytes derived from progenitors in the fetal liver[24,30,44,62,63]. This conclusion is consistent with analysis of expression of tissue macrophage-specific genes across an embryonic gene expression time course[65].

**The chromatin landscape of $Csf1r$.** Chromatin analysis and ChIP-seq data identifies at least four other $Csf1r$ enhancers that are active in mouse BMDM. Each is bound by JUN, STAT1, PU.1, and CEBPA, which also bind cooperatively to many other macrophage-associated enhancers (reviewed in ref. [9]). To extend these findings to other macrophage populations, we explored data from a previous study that compared the enhancer profiles of monocytes, microglia, and macrophages isolated from lung, liver, spleen, intestine, and peritoneal cavity[66]. Peaks of H3K4me1 methylation (indicative of poised enhancers) appear throughout the $Csf1r$ locus and extend upstream into the neighboring $Pdgfrb$ gene (Fig. 9a). This extended H3K4me1 signal is almost undetectable in neutrophils, which express $Csf1r$ mRNA and $Csf1r$ reporter genes at low levels but do not express surface CSF1R[11]. These additional elements lie outside the 3.5 kb promoter region, and intron containing FIRE, used in production of $Csf1r$-reporter genes (Fig. 9a). The ATAC-seq data revealed heterogeneity amongst monocyte–macrophage populations. FIRE is clearly a major peak of open chromatin detected in microglia and peritoneal macrophages, consistent with the impact of deletion of this

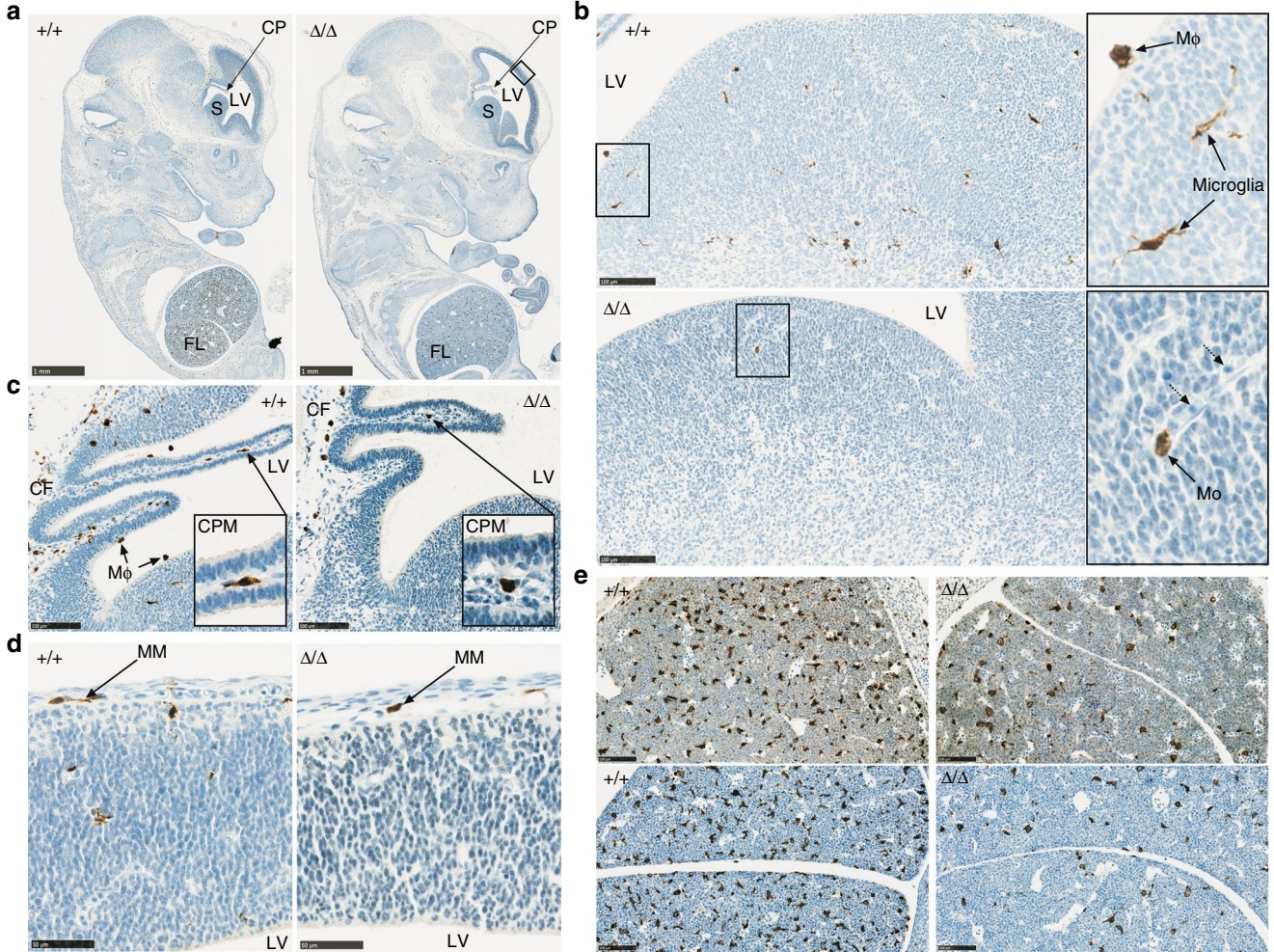

**Fig. 8** FIRE deletion results in a reduction of embryonic macrophages. **a** Formalin-fixed paraffin-embedded embryos (E13) were stained with antibodies against IBA1. Images are representative of seven embryos per genotype at E12, E12.5, and E13 and four repeat experiments. CP choroid plexus, LV lateral ventricle, S striatum, FL fetal liver. Black rectangle indicates the area magnified for panel **d**. Scale bar = 1 mm. **b** View of the striatum from E12.5 embryos highlighting IBA1+ microglia and ventricular macrophages (MΦ) in the control embryo and a monocyte (Mo) in the FIRE deficient embryo. Dotted arrows point to immature red blood cells in a blood vessel. Scale bar = 100 μm. **c** Images of the choroid plexus in the lateral ventricle (LV) of E12.5 embryos. CPM choroid plexus macrophage, MΦ ventricular macrophages, CF choroidal fissure. Scale bar = 100 μm. **d** IBA1+ meningeal macrophages (MM) in the forebrain (boxed area in panel **a**) of control and $Csf1r^{\Delta FIRE/\Delta FIRE}$ embryos at E12.5. Scale bar = 50 μm. **e** IBA1+ fetal liver monocytes at E12 and E13 in upper and lower panels, respectively. Scale bar = 100 μm

element (Fig. 9b). By contrast, the signal associated with FIRE is barely detectable in isolated lung and intestinal macrophages. Indeed, the promoter-associated ATAC-seq signal is barely detectable, consistent with relatively low expression of $Csf1r$ mRNA in these cells[35,67,68] and lack of phenotype observed in these tissues in $Csf1r^{\Delta FIRE/\Delta FIRE}$ mice[21]. The ATAC-seq data in Fig. 9b reveals multiple additional elements in cells that clear the majority of CSF1 from the circulation (notably in KC and splenic macrophages), that likely contribute to the redundancy of FIRE and to the high level of $Csf1r$ expression in these cell types[11]. One of these elements includes the distal promoter element (DPE). Located 5′ of the macrophage transcription start site, the DPE also has tissue-specific functions[64]. It includes a separate transcription start site cluster used by osteoclasts[69]. Expression of an amplified $Csf1r$-ECFP (MacBlue) reporter transgene, in which the DPE was deleted from the promoter, was undetectable in most tissue macrophages[64]. The exceptions that retained reporter gene expression were those populations impacted in the $Csf1r^{\Delta FIRE/\Delta FIRE}$ mouse; embryonic macrophages, BM progenitors and monocytes, microglia, Langerhans cells, and peritoneal

macrophages. Therefore, we conclude that FIRE and the DPE interact to control expression of $Csf1r$ in embryonic macrophages and in some tissue-resident macrophage populations but that both are partly redundant.

## Discussion

We show that the regulatory element, FIRE, is not required to support $Csf1r$ transcription in all macrophage populations. FIRE is essential for the generation or maintenance of specific CSF1R-dependent tissue macrophages, including those of the embryo, the brain (microglia), skin (Langerhans cells), peritoneum, heart, and kidney but redundant for expression of $Csf1r$ mRNA in adult mononuclear phagocyte populations in the intestine and lung. FIRE is not required for the development of CSF1R-dependent macrophages in other locations (i.e. dermis, spleen, liver, and bone). The microglial population of the brain is established early in development from yolk sac-derived progenitors. Apart from a transient postnatal monocyte influx, microglia are maintained by self-renewal in a CSF1R-dependent manner[70]. The loss of IBA1+ cells in $Csf1r^{\Delta FIRE/\Delta FIRE}$ embryos (Fig. 8) indicates that

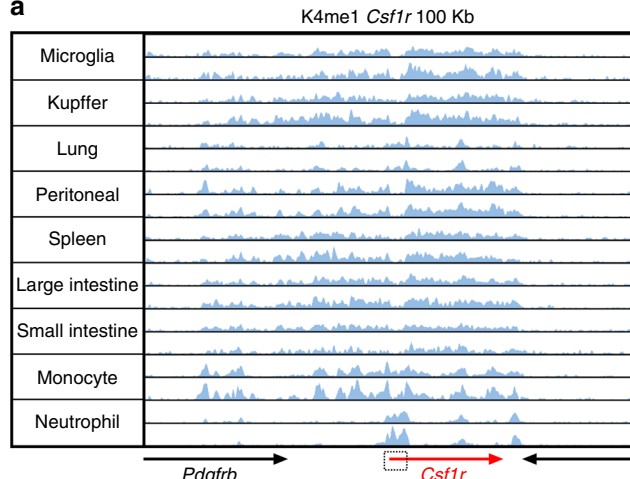

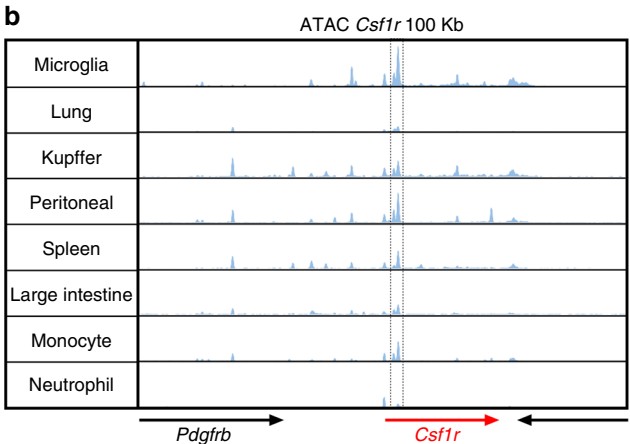

**Fig. 9** Chromatin landscape of the mouse *Csf1r* locus. ChIP-seq and ATAC-seq data for the 100 kb window surrounding mouse *Csf1r* for each of the isolated myeloid cell populations shown was extracted from the primary data generated in Lavin et al.[66]. **a** ChIP-seq data showing the location of poised enhancers (H3K4me1). The boxed area indicates the 3.5 kb promoter region used in the production of *Csf1r* reporter genes. **b** ATAC-seq data showing the locations of open chromatin. The dotted lines surround the double peak in the second intron; the more prominent peak is FIRE

microglial development fails early in embryogenesis. This is consistent with the apparent failure of macrophage production from *Csf1r*$^{\Delta FIRE/\Delta FIRE}$ EB in the E14 ESC model (Fig. 1f).

The relative independence of the large majority of tissue-resident macrophages on the enhancer activity of FIRE begs the question of why this element is so highly conserved across evolution. Enhancer redundancy (i.e. the presence of shadow enhancers), is widespread in eukaryotic developmental systems[71]. Nevertheless, redundancy is often incomplete and the non-overlapping functions presumably explain why multiple enhancers within a locus are conserved across evolution[71]. The core element within FIRE, conserved from reptiles to humans, contains binding sites for AP1 and PU.1 that are essential for enhancer activity[14]. Colocalization of AP1 and PU.1 motifs is a feature of many mouse macrophage-specific constitutive enhancers[72]. The non-redundant role of FIRE in *Csf1r* transcription in only a subset of macrophages may be a consequence of tissue-specific expression of transcription factors that bind selectively to this region of the *Csf1r* locus. Candidate regulators have been identified in many CSF1R-dependent macrophage populations.

For example, microglia are uniquely dependent upon specific transcriptional regulators encoded by *Sall1*[73] and *Irf8*[74]. Whilst multiple open chromatin sites are bound by PU.1, CEBPA, and other transcription factors in BMDM, IRF8 bound strongly and exclusively to FIRE (reviewed in ref. [9]). *Irf8* mRNA is co-expressed with *Csf1r* in the earliest committed monocyte progenitors[75] and upregulated in erythro-myeloid progenitors in parallel with *Csf1r* during early embryonic development[67]. Renal macrophages, which are also lost in *Csf1r*$^{\Delta FIRE/\Delta FIRE}$ mice, along with blood monocytes, share high gene expression of *Irf8* with microglia[67]. RUNX1 is another transcription factor that interacts with FIRE in both mouse and human macrophages and controls CSF1-dependent macrophage growth[76]. Accordingly, *Runx1* mRNA is expressed in the earliest progenitors of CSF1R-positive microglia[44] and is amongst the transcription factors down-regulated in the hippocampus in *Csf1r*$^{\Delta FIRE/\Delta FIRE}$ mice (Supplementary Data 1).

CSF1R deficiency causes embryonic or perinatal death in most mouse strains and the few surviving *Csf1r* mutant mice have numerous developmental abnormalities[77]. The *Csf1r*$^{\Delta FIRE/\Delta FIRE}$ mice that are born survive and thrive. Aside from survival, other major differences between the pleiotropic impacts of *Csf1r*$^{\Delta FIRE/\Delta FIRE}$ and *Csf1r*$^{-/-}$ mutations[4,77] are clearly related to the selective retention of most *Csf1r*-dependent macrophage populations in mice with the FIRE deletion. The expression profiling of the hippocampus supports the view that trophic and phagocytic functions attributed to microglia can be compensated in part by other cells. Notably, neuronal development, including expression of markers such as *Cux1*, is unaffected by the complete loss of *Csf1r* expression. This observation argues strongly against a functional role for *Csf1r* in neuronal progenitors[58] and instead supports that *Csf1r* is expressed and functional only in microglia[19]. So, we suggest that the reported developmental and homeostatic roles of *Csf1r*-dependent macrophages, including microglia and those of the epidermis, peritoneum, kidney, and heart are at least partly redundant. By extension, the loss of most peripheral macrophages and the associated systemic impacts of that loss must contribute to the severe brain developmental abnormality observed in *Csf1r*$^{-/-}$ mice[19,78]. Severe postnatal growth retardation is a shared feature of *Csf1* and *Csf1r* mutant mice and rats[4,5] and is a consequence of impacts on the growth hormone (GH)/insulin-like growth factor 1 (IGF1) axis (reviewed in ref. [79]). By contrast to the *Csf1r*-deficient mice, liver macrophages develop normally in *Csf1r*$^{\Delta FIRE/\Delta FIRE}$ animals (Fig. 4a and Supplementary Fig. 4a) and somatic growth, as well as the male–female sexual dimorphism in body weight gain, is unaffected by the mutation. In *Csf1* mutant rats, there was no post-natal surge in circulating IGF1[79]) and the *Csf1r* mutation in rats reduced expression of both *Ghr* and *Igf1* mRNA in the liver[5]. Circulating IGF1 in turn crosses the blood brain barrier and regulates many aspects of neuronal health (reviewed in ref. [80]).

In overview, our objective in this study was to investigate the role of a conserved enhancer in a key macrophage-expressed gene. We have confirmed the functional importance of FIRE in specific macrophage populations. In the process, we have generated a unique mouse model in which to identify non-redundant functions of specific resident tissue macrophages, especially those of the brain, heart, skin, kidney, and peritoneum, in mice that are otherwise healthy and fertile.

## Methods

**CRISPR design and plasmid preparation**. gRNAs were designed to complement the flanking regions of the FIRE element of intron 2 (from 2646 to 3015 bp downstream of the end of exon 2; NCBI Reference Sequence: NM_001037859.2) using the online tool at http://crispr.mit.edu. Table 1 shows the guides designed, the corresponding protospacer-adjacent motif (PAM) and the complementary

### Table 1 CRISPR guide RNAs

| Guide | Top BbSI-flanking sequence | Guide sequence | Bottom BbSI-flanking sequence | PAM |
|---|---|---|---|---|
| US1-top | CACC | 5' GAGTCCCTCAGTGTGTGAGA | – | AGG |
| US1-bottom | – | 3' TCTCACACACTGAGGGACTC | CAAA | – |
| DS1-top | CACC | 5' GGGATGACACAACGGTTTCC | – | TGG |
| DS1-bottom | – | 3' GGAAACCGTTGTGTCATCCC | CAAA | – |
| DS2-top | CACC | 5' CAATGAGTCTGTACTGGAGC | – | AGG |
| DS2-bottom | – | 3' GCTCCAGTACAGACTCATTG | CAAA | – |

*US* upstream FIRE, *DS* downstream FIRE, *PAM* protospacer-adjacent motif

BbSI-overhang sequences. Top and bottom oligonucleotides corresponding to each gRNA were aligned, phosphorylated, and cloned into the pSpCas9(BB)-2A-GFP (Px458) plasmid (Addgene), and transformed into competent DH5α *E. coli*. Following verification of gRNAs insertion by Sanger sequencing, plasmids were purified using endotoxin-free MaxiPrep (QIAGEN).

**Cell culture**. RAW 264.7 macrophages (ATCC® TIB-71™) were cultured on Sterilin-plastic dishes in RPMI 1640 medium (Sigma) supplemented with 10% endotoxin-free FBS (GE Healthcare), 2 mM GlutaMAX (Invitrogen), 25 U/mL of penicillin, and 25 μg/mL of streptomycin (Gibco). Cells were incubated at 37 °C in a 5% $CO_2$ humidified incubator. E14 (129P2/OlaHsd) mouse ESC (ATCC® CRL-1821™) were cultured on porcine gelatin-coated (Sigma) flasks in GMEM (Invitrogen) supplemented with 10% ESC-tested FBS (Hyclone), 2 mM GlutaMAX (Invitrogen), 1X MEM non-essential amino acids, 1 mM sodium pyruvate, 0.1 mM 2-mercaptoethanol (Gibco), and 100 U/mL leukemia inhibitory factor (LIF, Merck), at 37 °C in a 5% $CO_2$ humidified incubator.

**Generation of CRISPR/Cas9 deletions in vitro**. E14 cells were passaged twice and plated at $8 \times 10^4$ cells/cm² for 24 h prior to transfection. Cells were transfected with a total of 2 μg DNA (i.e. pairs of Cas9-2A-EGFP vectors, each expressing single gRNAs: US1+DS1 or DS2) using the Amaxa P3 Primary cell 4D-Nucleofector X Kit (Lonza) and incubated with supplemented GMEM for 36 h. Cells were then resuspended in Dulbecco's phosphate buffered saline (DPBS, Sigma), diluted in sterile Baxter water (TPS Healthcare/Baxter), supplemented with 2% FBS and kept at 4 °C. The GFP⁺ cells were collected with a FACS Aria IIIu cell sorter (BD). To generate single cell clones of CRISPR-modified E14 cells, sorted pools of GFP⁺ cells were plated on 0.1% gelatin-coated 100 mm culture dishes at a density of 18 cells/cm² and single colonies picked approximately 2 weeks later. RAW 264.7 cells ($5 \times 10^6$) were resuspended in 250 μL of culture medium containing 10 μL of DPBS ± 20 μg DNA (as pairs of FIRE-targeting Cas9-2A-EGFP vectors, as specified above) and incubated at RT for 10 min. Electroporation was performed in 0.4 cm electroporation cuvettes (BioRad) using the BioRad Gene Pulser II (BioRad), at 320 V and a capacitance of 950 μF. Cells were cultured for 36 h and GFP⁺ cells were single-cell sorted into polystyrene flat-bottom 96-well plates, using a FACS Aria IIIu cell sorter (BD).

**PCR analysis and sequencing of transfected E14 ESC and RAW 264.7 cells**. Isolation of DNA from sorted pools of GFP⁺ cells (either E14 ESC or RAW 264.7 cells), or from cells cultured in plates other than 96-well plates was performed with the QIAGEN DNeasy Blood and Tissue Kit (QIAGEN), according to the manufacturer's instructions. Genomic DNA from single-cell-derived clones cultured in 96-well plates was isolated by adding a 1:1 mix of DirectPCR Lysis Reagent (Viagen) and nuclease-free water (Invitrogen) supplemented with 20 μg of Proteinase K (QIAGEN), per well. Cells were incubated in a heated orbital shaker at 200 rpm and 55 °C, for 12 h and Proteinase K was inactivated at 85 °C for 45 min. Conventional PCR was performed using 50 ng DNA per reaction, Taq DNA polymerase (Invitrogen) and the primers F-5′: GCTGCCCTGTCACTGTGTA and R-5′: TCGTTTCCCATCCCAGGA, at an annealing temperature of 55 °C. PCR products were run by electrophoresis to obtain a *Csf1r*⁺/⁺ of 925 bp, or *Csf1r* ᐞFIRE/ᐞFIRE amplicons of either 485 bp (CRISPR pair: US1 and DS1) or 507 bp (CRISPR pair: US1 and DS2). PCR amplicons were separated by standard electrophoresis, purified with the QIAquick PCR purification kit (QIAGEN) and then sequenced using the oligonucleotides listed in Supplementary Table 1.

**DNA isolation and genotyping**. Mouse ear biopsies were digested overnight at 50 °C, in mouse biopsy buffer containing 100 mM Tris–HCl (Sigma) pH 8.5, 0.2% SDS (Ambion), 200 mM NaCl (Sigma), 5 mM EDTA (Ambion), and 160 μg Proteinase K (QIAGEN). Digested tissues were centrifuged at 12,000×g for 15 min at 4 °C and DNA was precipitated through the isopropanol–ethanol method. DNA pellets were dried at RT, resuspended in nuclease-free water and quantified using a Nanodrop™ 1000 (Thermo Fisher Scientific). PCR amplification was performed as described above and products were purified using the QIAquick PCR purification kit (QIAGEN) and sequenced using the oligonucleotides listed in Supplementary Table 1.

**Phagocytosis assay**. RAW 264.7 cells were cultured in high glucose DMEM (Gibco) supplemented with 10% endotoxin-free FBS (GE Healthcare) and 1 mM GlutaMAX (Life Technologies). Cells were incubated at 37 °C for 2 h in the presence of pHrodo® Red *E. coli* BioParticles® (Thermofisher) at a ratio of 100 μg/10⁶ cells. Negative control samples were incubated on ice. Cells were analyzed using the BD LSR Fortessa flow cytometer. Dead cells were identified and excluded based upon DAPI uptake.

**Flow cytometry**. Cells were analyzed by flow cytometry using standard procedures. Inhibition of non-specific binding was achieved by incubating the cells for 30 min on ice in purified anti-CD16/32 (1 μg/mL, BioLegend, 101302) or hybridoma 2.4G2 supernatant. All antibodies are listed in Supplementary Table 2. Where indicated, cells were fixed and permeabilized prior to staining using the Leucoperm™ kit (BioRad). Blood was processed using the Uti-Lyse reagent (Agilent). Data analysis was performed in FlowJo® v10.0 (Tree Star). Gates were determined with isotype controls or FMO for all experiments.

**Differentiation of ESC-derived macrophages**. E14 ESCs were harvested and plated at a density of $5 \times 10^6$ cells per 50 mm Sterilin-plastic dishes, using culture medium without LIF. After 4 days, cell debris was removed using 70 μm cell strainers. Fresh culture medium containing 50 ng/mL of rhCSF1 (a gift from Chiron, USA) and 10 ng/mL of mouse recombinant interleukin 3 (IL3, Invitrogen) was added. After 7 days the culture medium was replaced to contain 100 ng/mL of rhCSF1 for 10 days, without IL3.

**Generation of Csf1r$^{\Delta FIRE/\Delta FIRE}$ mice**. Ethical approval was obtained from The Roslin Institute's and The University of Edinburgh's Protocols and Ethics Committees, under the authority of a UK Home Office Project License under the regulations of the Animals (Scientific Procedures) Act 1986. Mice were bred and housed under specific pathogen-free conditions. *Csf1r*$^{\Delta FIRE/\Delta FIRE}$ mice were produced by pronuclear injection of oocytes with the plasmids encoding FIRE-CRISPRs US1 and DS2 (10 μg of each) and *Cas9* mRNA. Both donor and recipient females were B6CBAF1/J (JAX™, stock number 100011). Founders were then crossed to C57BL/6 mice and their offspring interbred.

**Micro CT**. Micro CT was performed on formalin-fixed, paraffin-embedded decalcified femurs. Specimens were held in 1% agarose in a 20 mL universal tube before mounting in a Skyscan 1172 desktop micro CT (Bruker). The tubes were then scanned through 360° using a step of 0.28° between exposures. A voxel resolution of 6.03 μm was obtained using the following control settings: 54 kV source voltage, 185 μA source current with an exposure time of 1767 ms. A 0.5 mm aluminum filter and two frame averaging were used to optimize the scan. After scanning, the data was reconstructed using Skyscan software NRecon v1.6.9 (Bruker). The reconstruction thresholding window was optimized to encapsulate the target image, with the same values used for all bones. Reconstructed bones were oriented around their vertical axis using DataViewer v1.5.2.4 (Bruker). The trabecular region of bone 200 μm thick starting 100 μm proximal to the primary spongiosa was identified, and volumetric analysis to determine the microarchitecture of this region was performed using CTAn v1.16.4.

**Histology and immunohistochemistry**. Tissues were fixed in 10% neutral buffered formalin and processed into paraffin using standard procedures. For examination of tissue architecture, sections were stained with hematoxylin and eosin (H&E) or Luxol fast blue. Femurs and tails were decalcified in EDTA. Osteoclasts were detected by staining sections with the tartrate-resistant acid phosphatase (TRAP) kit (Sigma), according to instructions.

Goblet cells were stained by treating sections of small intestine with 1% periodic acid for 7 min and incubating them in Schiff's reagent for 15 min, with washes in water between steps; followed by counterstaining for 15 s with Hematoxylin.

For immunohistochemistry, antigen retrieval was performed with Proteinase K for 5 min at 37 °C prior to staining. Tissue-resident macrophages in spleen and liver sections were detected by staining with the rat anti-mouse F4/80 monoclonal antibody (1:600; AbD Serotec, MCA497G). For staining of adult brains, primary antibodies were applied overnight at 4 °C in a humid chamber. These included

chicken anti-GFAP (1:500; Cambridge Bioscience, 829401) and rabbit anti-SOX9 (1:500; Millipore, AB5535). Following three washes in PBS, fluorescently conjugated secondary antibodies were applied (1:500; Life Technologies) for 2 h at RT, then slides were counterstained with DAPI for 10 min and mounted with Fluoromount G (Cambridge Bioscience). For isolation of epidermal sheets, dissected tail skin was incubated in 2 mg/mL Dispase II (Sigma) at 37 °C for 20 min. Epidermal skin samples were fixed overnight in 4% PFA at 4 °C. After washing with TBS, samples were incubated for 1 h at RT in permeabilization buffer (PB)—TBS containing 0.25% gelatin from cold water fish skin (Sigma), 0.5% Triton X100 (Sigma), and 5% heat-treated goat serum (Sigma). Epidermal samples were then incubated in PB containing rat anti-mouse MHCII (Clone 2G9, 1:100, BD, 553621) and 1:30 goat anti-human Langerin (1:30, Thermo Fisher Scientific, PA5-47250) overnight at 4 °C. Samples were washed five times for 1 h at RT in TBS containing 0.02% Tween 20 (Fisher Scientific) (TBST) followed by incubation overnight at RT with anti-goat and anti-rat Alexa Fluour secondary antibodies (Life Technologies) diluted 1:500 in PB. After washing four times for 1 h in TBST, epidermal sheets were counterstained with DAPI (Sigma) and mounted in Prolong Gold (Life Technologies).

Frozen sections of 4% PFA perfused adult brains were processed and stained with rabbit anti-IBA1 (1:500, Wako, 019-19741), rabbit anti-P2RY12 (Sigma, 1:125, HPA013796), and Alexa Fluor® conjugated secondary antibodies using standard procedures.

Dura mater were removed into ice-cold PBS, fixed for 1 h in 4% PFA, treated for 30 min at 37 °C in PBS containing 20 mM EDTA and then blocked in 3% BSA, 0.3% Triton X100 in PBS for 30 min. Whole mount staining steps were performed in blocking buffer containing rabbit anti-IBA1 overnight (1:1000, Wako, 019-19741), goat anti-rabbit Alexa Fluor® 594 1 h (1:400, Life Technologies, A-11012), Alexa Fluor® 647 anti-mouse CD169 1 h (Clone Siglec-1, 1:200, BioLegend, 142408), with washes in buffer in between incubations. Stained tissues were then mounted onto glass slides with mounting medium and cover-slipped. Slides were imaged via a LSM710 confocal microscope (Zeiss) using Zen software.

**Image acquisition and quantification**. Whole-slide bright field images were acquired using the NanoZoomer slide scanner. Image analysis was performed with the NDP.view software v2.4.26 (Hamamatsu) and with ImageJ v1.46h. For spleen and liver, 10 regions of interest (ROI) were exported per sample as jpg format from the NDP.view files. Signal was quantified from the whole area corresponding to each ROI (i.e. each jpg file). For analysis of small intestine, individual villi were set as a ROI in the jpg file derived from PAS-staining. Villus perimeter, as well as the area and the number of Goblet cells (PAS+ cells) were quantified. Measurement of villus length was performed in H&E images where 30 villi were measured per sample, from the mucosal base adjacent to the crypts to the apex, using the "ruler" function of the NDP.view v2.4.26 software. Entire brain tissue sections were imaged using a Zeiss AxioScan SlideScanner and visualized with Zen2 software. Epidermal tail sheets were imaged with a Zeiss LSM 710 confocal microscope and ZEN software. For astrocyte analysis the images were obtained by z-stacks using a Zeiss Axio Scan.Z1 (10× objective) and an Olympus 3i Spinning Disk confocal microscope (30× silicone objective) using SlideBook software. For astrocytes counts, tiff files from max projections were imported to Image J, and the images were converted to 8-bit fluorescence RGB tiff and thresholded. For nuclear counts, watershed segmentation on binary images was used to separate objects and then images were analyzed. Two fields of view were quantified and then the mean was calculated and converted to mm². To count the proportion of GFAP-positive cells, the number of GFAP+ cells was divided by the number of SOX9+ nuclei.

**Blood analysis**. Blood was isolated through cardiac puncture, using syringe and needle coated with 2% EDTA. Blood was collected into 2 mL EDTA tubes (BD Vacutainer®) containing 1.8 mg of EDTA per mL of blood. Total blood cell counts were performed using the ABX Pentra 60 hematology analyzer and differential counts of WBC subsets were performed on blood smears.

**BM isolation and cell culture**. BM was isolated by flushing femurs with RPMI containing 5 mM EDTA. For flow cytometry analysis, erythrocytes were not lysed to prevent cleavage of CSF1R. For macrophage differentiation, erythrocytes were lysed with RBC lysis buffer (BioLegend) and BM cells were cultured on Sterilin-plastic dishes in the presence of rhCSF1 (at 100 ng/mL, a gift from Chiron) for 7 days. Fresh media containing rhCSF1 was added on Day 4.

**Isolation of peritoneal cells**. Cells for flow cytometry analysis were isolated from mice by lavage of the peritoneal cavity with PBS. Cells were centrifuged at 400×g for 5 min at 4 °C prior to resuspension in PBS containing 2% endotoxin-free FBS (GE Healthcare).

**Preparation of organs for flow cytometry**. The spleen, liver, lung, heart, and kidneys were prepared for flow cytometry from non-perfused mice. Organs were removed, chopped finely, and digested in RPMI containing 0.625 mg/mL collagenase D (Roche), 0.85 mg/mL collagenase V (Sigma), 1 mg/mL dispase (Life Technologies), and 30 U/mL DNase (Roche) for 22 min (lung) or 45 min (other organs) in a shaking incubator at 37 °C. Organ preparations were passed through

100 μm cell strainers and centrifuged at 300×g for 5 min at 4 °C prior to resuspension in PBS containing 2% endotoxin-free FBS. Erythrocytes were then lysed using RBC lysis buffer (BioLegend).

**Isolation of intestinal cells**. Single cell suspensions of small and large intestines (colon) from non-perfused mice were prepared for flow cytometry. Intestines were opened longitudinally and cut into 2–3 cm segments in ice-cold PBS and intestinal contents removed by gentle shaking. Tissue segments were incubated at 37 °C for 20 min whilst shaking in RPMI containing 3% endotoxin-free FBS (GE Healthcare), 20 mM HEPES (Gibco), 5 mM EDTA (Sigma), 1 mM DTT (Promega), and 100 U/mL polymyxin B (Sigma). Segments were transferred to RPMI containing 2 mM EDTA and 20 mM HEPES and shaken by hand to ensure optimal dissociation of epithelial cells and lamina propria leukocytes. Tissues were minced with scissors and digested at 37 °C for 30 min, whilst shaking in RPMI containing 20 mM HEPES, 0.425 mg/mL Collagenase V (Sigma), 0.625 mg/mL Collagenase D (Roche), 1 mg/mL Dispase (Gibco), and 30 μg/mL DNase (Roche). Cell suspensions were passed through 70 μm then 40 μm cell strainers in RPMI containing 10% endotoxin-free FBS and 100 U/mL polymyxin B. After centrifugation at 400×g at 4 °C for 5 min, including a wash step, cells were resuspended in PBS containing 2% FBS for flow cytometry.

**Myelin depletion of brains and isolation of cells**. Myelin-depleted brain suspensions were prepared from saline perfused mice for flow cytometry. Whole brains were finely minced with scissors in ice-cold Hank's balanced salt solution (HBSS, without calcium or magnesium) and centrifuged at 400×g for 5 min at 4 °C. Minced brains were then digested in HBSS containing 50 U/mL Collagenase IV (Gibco), 100 μg/mL Nα-Tosyl-L-lysine chloromethyl ketone hydrochloride (Sigma), 5 U/mL DNase I (Roche) and 8.5 U/mL Dispase (Gibco) for 1 h at 37 °C whilst shaking. The digested tissue was homogenized in a glass Dounce tissue grinder for 20 passes and an equal volume of HBSS containing 10% endotoxin-free FBS was then added. After centrifugation at 400×g for 5 min at 4 °C, the cell pellet was resuspended in 16 mL 35% Percoll in HBSS, overlaid with 10 mL HBSS and incubated on ice for 5 min. The gradient was centrifuged at 800×g for 45 min at 4 °C with no brake. The cell pellets were then resuspended in PBS containing 2% FBS for flow cytometry.

**RNA isolation and microarrays**. For RNA isolation, tissues from saline perfused mice were snap frozen, and subsequently disrupted in the Precellys24 Homogenizer® (Bertin Instruments). RNA isolation was performed using the RNeasy Plus Mini kit (QIAGEN). Library preparation and hybridization to the Affymetrix Mouse Gene 1.0 ST array was performed by Edinburgh Genomics, University of Edinburgh. CEL files were RMA normalized and annotated in R/Bioconductor. Analysis of differential expression was performed using the Affymetrix Transcription Analysis Console (Thermo Fisher).

**Analysis of hippocampi microarray data**. These data were interrogated with sets of genes whose expression is highly enriched in specific cell types of the brain (astrocytes, oligodendrocytes, microglia, neurons, brain endothelial cells). These sets were curated from published RNA-seq data (GEO, accession number GSE73721) describing the transcriptome of astrocytes, oligodendrocytes, microglia, neurons, and brain endothelial cells[60]. The inclusion criteria consisted of (a) being expressed at least five-fold higher in the cell type of interest than all other cell types and (b) being expressed at least 50 FPKM to enable robust detection in the mixed cell type environment of a whole hippocampus.

**cDNA synthesis and quantitative real-time PCR**. Complementary DNA (cDNA) was synthesized from RNA using the SuperScript III. First-Strand Synthesis System and remaining template was removed by incubation with RNase H (Invitrogen). Oligonucleotides were designed in Primer3 v4.0.0 (available at http://bioinfo.ut.ee/primer3/). Real-time qPCR reactions were performed using the Fast SYBR green master mix and the 7500 Fast System (Applied Biosystems). Analysis was performed with 7500 software v2.0.6 (Applied Biosystems, Life Technologies Corporation). For analysis of relative changes in gene expression, data were normalized according to the $2^{-\Delta\Delta CT}$ method. Hypoxanthine guanine phosphoribosyl transferase (Hprt) was used as a loading control/housekeeping gene. Oligonucleotide sequences are listed in Supplementary Table 3.

**Reporting summary**. Further information on research design is available in the Nature Research Reporting Summary linked to this article.

## Data availability

The authors declare that all data supporting the findings in this study are available within the article and its Supplementary Information files or from the corresponding authors on reasonable request. The RNA expression datasets are publicly available in the gene expression omnibus (GEO; https://www.ncbi.nlm.nih.gov/gds) via the accession number GSE108207. Published ATAC-seq and ChIP-seq data used in this manuscript is available via GEO accession number GSE63341. The RNA-seq data used to curate cell type specific

genes in the brain is available via GEO accession number GSE73721. The data used to identify microglia-enriched genes versus macrophage genes is available via GEO accession number GSE48579. All other data contained within this manuscript is located in the Source Data excel file.

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

## Acknowledgements

This work was supported by the medical research council (MRC) UK grant MR/M019969/1. R.R. was supported by a doctoral scholarship (application number: 314413, file number: 218819) granted by CONACyT Nuevo Leon—I2T2, Mexico. J.A. was supported by Wellcome Trust grant 201531/Z/16/Z. D.A.H. and K.M.S. are supported by The Mater Foundation. P.H. was supported by the biotechnology and biological sciences research council (BBSRC) grant BB/P013732/1. D.D.O. was supported by MRC grant MR/M010341/1. C.P. is supported by the Simons Foundation Autism Research Initiative. Flow cytometry data was generated with support from the Queen's Medical Research Institute (QMRI) and Roslin Institute Flow Cytometry and cell sorting facility, University of Edinburgh. Histology data was generated with support from Easter Bush Pathology and Shared University Research Facilities at the University of Edinburgh. We thank the assistance provided by animal technicians at the University of Edinburgh, particularly Christine Marshall.

## Author contributions

D.A.H. and C.P. conceived the study. P.H. and D.D.O. supervised the CRISPR/Cas9 work. R.R., D.A.H. and C.P. wrote the manuscript. K.G., E.W., V.E.M., D.A.D.M, J.P., N.A.M. and K.M.S. edited the manuscript. R.R., A.R., D.D.O., L.L., K.G., E.W., M.C., I.G., A.S., Z.M.L., J.A., B.B., I.M., H.D., R.J.L., D.A.D.M., J.D.G., R.W., S.J.J., M.B. and C.P. performed experiments. E.D. and I.A. provided ChIP-seq and ATAC-seq data. R.R., G.E.H., K.M.S., D.A.H. and C.P. analyzed data.

## Additional information

**Competing interests:** The authors declare no competing interests.

