## [Peer Review File · Nature Communications]

Editorial Note: This manuscript has been previously reviewed at another journal that is not operating a transparent peer review scheme. This document only contains reviewer comments and rebuttal letters for versions considered at Nature Communications .

Reviewers' comments:

Reviewer #1 (Microglia, macrophage; original Nature Immunology referee #1)(Remarks to the Author):

The authors have responded to most of my concerns and the manuscript is significantly strengthened. This is a well-performed study with a very interesting and provocative finding that provides an interesting twist to our understanding of the macrophage compartment. I support publication as is

minor remaining issues

Ly6C- monocytes are not 'anti-inflammatory', this should be corrected in the introduction

I would suggest that the ATAC and CHIPseq analysis the authors refer to in the beginning of their discussion is taken to the results part.

The genetic background should be clearly stated in the Material & Methods section (as in the response to the reviewer). Intercrossing of littermates does there not help. The mixed background probably also kept the authors from performing the obvious experiment, i.e. a mixed BM transfer. this could be revealing.

Reviewer #2 (Mucosal immunity, macrophage; original Nature Immunology referee #3)(Remarks to the Author):

The authors have included additional experimental findings and modified many of the problems raised in the original manuscript, with the overall effect of improving the readability and interpretation of the study. However important aspects remain unsatisfactory.

1) The continued absence of detailed phenotypic and/or transcriptional information on defined subsets of macrophages in the intestine and brain is simply unacceptable. The authors' replies to this criticism are ingenuous and not helpful. For instance, while the comments about enzymatic digestion causing artifacts do indicate the need for caution when using such methods, they are now standard for the community, who will not be convinced by the use of total tissue RNA as a means of identifying cell populations or processes. It is irrelevant how many macrophages are present in the tissues, especially when many of the markers referred to are not specific to macrophages and can be expressed by other cells present in eg the intestine in almost as high numbers (eg F4/80 is found on eosinophils; CD11b on multiple myeloid cells). Even more tellingly, the authors themselves use enzymatic digests (without methods being stated) for all the other tissues described in the study, now conveniently oblivious to their concerns about the reservations raised. Without appropriate cellular detail, no conclusions can be made about macrophage populations in tissues where heterogeneity exists and it is surprising that the authors continue to assert that this is not required.

2) These issues are compounded by the fact that to find support for the authors' interpretations of these tissue RNA data, one is referred to Supplementary Tables which still lack appropriate annotation and remain indigestible. For instance, Supplementary Table 1 consists of >9000 pages with the gene probes listed without any annotation to assist the reader. Supplementary Table 2 is at least shorter and genes are named, but no relevant information is provided to coincide with the material discussed in the Results.

3) The more extensive gating information provided is welcome and considerably assists interpretation. However in the copy I received, the grey scale flow cytometry plots were extremely pale, making populations and annotations very difficult to see.

- 4) The lack of a biological effect of FIRE deletion on populations in which CSF-1R expression is abolished remains a confusing finding. The authors infer that this may be the case on some cells which normally do not express high levels of CSF-1R and/or the use of alternative promoter/enhancer elements. In the latter case, the authors discuss new information obtained in collaboration with Ido Amit. However if this material is considered sufficiently important to allow the current findings to be interpreted properly, it should not be cited as Supplementary data in the Discussion, but included in the Results. It also has to be noted that not all the authors' comments on the discrepancy between CSF-1R expression and effects of FIRE are consistent. For instance, on p27 it is stated that "FIRE was the dominant peak of open chromatin detected in microglia, peritoneal macrophages and blood monocytes, consistent with the impact of deletion of this element", despite the fact that blood monocytes behave completely differently from the other populations in the absence of FIRE in vivo.
- 5) On a similar note, if other growth factors in bone marrow may compensate for the loss of CSF-1R expression in FIRE^{-/-} mice (p26), why do these not also function in mice lacking CSF-1 or CSF-1R?
- 6) As noted above, no methods are provided for the enzymatic isolation of tissue derived cells, nor is it stated whether organs such as the lung and liver were perfused to ensure distinction between cells in the bloodstream and parenchyma.
- 7) In Supplementary Figure 5, it appears that CD11c expression may be reduced on interstitial lung macrophages from FIRE^{-/-} mice. Is this consistent?
- 8) In Figure 4, why were cDC2 in liver identified amongst a population of F4/80 expressing cells?
- 9) The comments on p18 about the normal populations of macrophages seen in the ovary and testis in FIRE^{-/-} mice being "consistent" with their essential role in the functions of these organs are hardly commensurate with the findings elsewhere in the paper from organs where the effects of FIRE deletion and macrophage numbers do not correlate with the known roles of macrophages in these organs (eg brain, intestine).
- 10) While improved, the Discussion remains long and discursive, especially with reference to the information on regulation of CSF-1R transcription when including information that should be in the Results (see above).
- 11) Some direct experimental evidence for the mechanisms speculated on as explanations for the discrepancies between the phenotype of FIRE^{-/-}, CSF-1^{-/-} and CSF-1R^{-/-} mice would have seemed appropriate if the new mouse model is to be useful.

Reviewers' comments:

Reviewer #1 (Microglia, macrophage; original Nature Immunology referee #1)(Remarks to the Author):

The authors have responded to most of my concerns and the manuscript is significantly strengthened. This is a well-performed study with a very interesting and provocative finding that provides an interesting twist to our understanding of the macrophage compartment.

I support publication as is. minor remaining issues

Ly6C- monocytes are not 'anti-inflammatory', this should be corrected in the introduction

The text has been corrected.

I would suggest that the ATAC and CHIPseq analysis the authors refer to in the beginning of their discussion is taken to the results part.

The data describing the chromatin landscape of *Csf1r* has been moved to the results section (new Figure 9).

The genetic background should be clearly stated in the Material & Methods section (as in the response to the reviewer). Intercrossing of littermates does there not help. The mixed background probably also kept the authors from performing the obvious experiment, i.e. a mixed BM transfer. this could be revealing.

Additional information about the genetic background is now included in the Material and Methods.

Reviewer #2 (Mucosal immunity, macrophage; original Nature Immunology referee #3)(Remarks to the Author):

The authors have included additional experimental findings and modified many of the problems raised in the original manuscript, with the overall effect of improving the readability and interpretation of the study. However important aspects remain unsatisfactory.

1) The continued absence of detailed phenotypic and/or transcriptional information on defined subsets of macrophages in the intestine and brain is simply unacceptable. The authors' replies to this criticism are ingenuous and not helpful. For instance, while the comments about enzymatic digestion causing artifacts do indicate the need for caution when using such methods, they are now standard for the community, who will not be convinced by the use of total tissue RNA as a means of identifying cell populations or processes. It is irrelevant how many macrophages are present in the tissues, especially when many of the markers referred to are not specific to macrophages and can be expressed by other cells present in eg the intestine in almost as high numbers (eg F4/80 is found on eosinophils; CD11b on multiple myeloid cells). Even more tellingly, the authors themselves use enzymatic digests (without methods being stated) for all the other tissues described in the study, now conveniently oblivious to their concerns about the reservations raised. Without appropriate cellular detail, no conclusions can be made about macrophage populations in tissues where heterogeneity exists and it is surprising that the authors continue to assert that this is not required.

We have now included additional IHC data in Figure 7 showing that FIRE deficient mouse brains retain meningeal, choroid plexus and perivascular macrophages, while microglia are absent. These data, alongside the tissue disaggregation confirming the loss of microglia, is complemented by the expression data on whole brain. The expression data both confirms the selective loss of microglia, and demonstrates that there is no impact of the lack of microglia on expression of transcripts associated with other cell types.

The new Supplementary Figure 6 contains flow cytometry data of both the small and large intestines and demonstrates that no macrophage populations are affected in the absence of FIRE. Furthermore, despite the lack of CSF1R on monocytes, there is no change in monocyte populations in the lamina propria isolated from the FIRE mutant mice. The methods sections in both the main article and supplementary data contain all information about enzymatic digests.

Whilst we have added these data, we do not believe that the expression data is uninformative or redundant. It demonstrates clearly that neither *Csf1r*, nor any known macrophage-associated transcript, is significantly altered in expression in the intestine in the mutant mice. This is clearly quite different from the brain.

2) These issues are compounded by the fact that to find support for the authors' interpretations of these tissue RNA data, one is referred to Supplementary Tables which still lack appropriate annotation and remain indigestible. For instance, Supplementary Table 1 consists of >9000 pages with the gene probes listed without any annotation to assist the reader. Supplementary Table 2 is at least shorter and genes are named, but no relevant information is provided to coincide with the material discussed in the Results.

We had analyzed the array data correctly; however, it was uploaded incorrectly during the resubmission process. This has been rectified upon resubmission.

3) The more extensive gating information provided is welcome and considerably assists interpretation. However in the copy I received, the grey scale flow cytometry plots were extremely pale, making populations and annotations very difficult to see.

All contoured flow cytometry plots have been updated to 2% contour with outliers (in FlowJo), which makes the plots easier to visualise.

4) The lack of a biological effect of FIRE deletion on populations in which CSF-1R expression is abolished remains a confusing finding. The authors infer that this may be the case on some cells which normally do not express high levels of CSF-1R and/or the use of alternative promoter/enhancer elements. In the latter case, the authors discuss new information obtained in collaboration with Ido Amit. However if this material is considered sufficiently important to allow the current findings to be interpreted properly, it should not be cited as Supplementary data in the Discussion, but included in the Results.

The data describing the chromatin landscape of *Csf1r* have been moved to the results section (new Figure 9).

It also has to be noted that not all the authors' comments on the discrepancy between CSF-1R expression and effects of FIRE are consistent. For instance, on p27 it is stated that "FIRE was the dominant peak of open chromatin detected in microglia, peritoneal macrophages and blood monocytes, consistent with the impact of deletion of this element", despite the fact that blood monocytes behave completely differently from the other populations in the absence of FIRE in vivo.

We agree with this comment and the relevant text has been updated (now in the Results section).

5) On a similar note, if other growth factors in bone marrow may compensate for the loss of CSF-1R expression in FIRE-/- mice (p26), why do these not also function in mice lacking CSF-1 or CSF-1R?

CSF1-deficient mice do recover with age, in part due to the actions of FLT3 ligand (PMID:11675341) and there is a second CSF1R ligand, IL34. *Csf1r* mutant mice lack osteoclasts, so they do not develop bone marrow. The FIRE -/- mice do not lack osteoclasts, so they develop bone marrow and can therefore sustain bone marrow progenitors. We have made this clearer in the text.

6) As noted above, no methods are provided for the enzymatic isolation of tissue derived cells, nor is it stated whether organs such as the lung and liver were perfused to ensure distinction between cells in the bloodstream and parenchyma.

Additional information has been included in the methods section.

7) In Supplementary Figure 5, it appears that CD11c expression may be reduced on interstitial lung macrophages from FIRE-/- mice. Is this consistent?

This is not consistent. The contour plots for the WT mouse have been replaced to be more representative.

8) In Figure 4, why were cDC2 in liver identified amongst a population of F4/80 expressing cells?

F4/80 is not a marker that clearly distinguishes cDC2 from macrophages. In the liver the putative cDC2 are F4/80 low (PMID:29440354).

9) The comments on p18 about the normal populations of macrophages seen in the ovary and testis in FIRE^{-/-} mice being "consistent" with their essential role in the functions of these organs are hardly commensurate with the findings elsewhere in the paper from organs where the effects of FIRE deletion and macrophage numbers do not correlate with the known roles of macrophages in these organs (eg brain, intestine).

We agree with this comment and the statement has been removed from the text.

10) While improved, the Discussion remains long and discursive, especially with reference to the information on regulation of CSF-1R transcription when including information that should be in the Results (see above).

We have modified the discussion in response to this comment.

11) Some direct experimental evidence for the mechanisms speculated on as explanations for the discrepancies between the phenotype of FIRE^{-/-}, CSF-1^{-/-} and CSF-1R^{-/-} mice would have seemed appropriate if the new mouse model is to be useful.

We do not agree that there are discrepancies between the phenotypes of these mouse models. The *Csf1^{op/op}* mutant lacks osteoclasts and many tissue macrophages. It is less severe than the *Csf1r* KO because of the second growth factor (IL34) and because CSF1 can cross the placenta. The *Csf1r*^{-/-} mouse lacks *Csf1r* completely and is severely osteoclast- and macrophage-deficient. Most mutant mice do not survive to weaning. We show the FIRE^{-/-} mouse is not *Csf1r*-deficient. It is a hypomorphic mutation. The selective effect of the mutation on *Csf1r* expression is explained by the fact that there are multiple additional enhancers in the *Csf1r* locus that are active in different macrophage populations. The selective phenotypic impact of the mutation on development is explained by the fact that osteoclasts and most tissue macrophage populations express *Csf1r* mRNA normally and are retained. The lack of any brain phenotype demonstrates unequivocally that microglia are not absolutely required for normal brain development.

REVIEWERS' COMMENTS:

Reviewer #2 (Remarks to the Author):

I thank the authors for the additional data and improved description/presentation of the results. Together these changes considerably help clarify the findings and interpretation. One minor comment:

1) From the analysis of intestinal cells shown in Supplementary Figure 6, it looks as if there may be decreases in total CD45+ cells and MHCII+ macrophages in the FIRE-/- mice. Were these changes significant?

REVIEWERS' COMMENTS:

Reviewer #2 (Remarks to the Author):

I thank the authors for the additional data and improved description/presentation of the results. Together these changes considerably help clarify the findings and interpretation. One minor comment:

1) From the analysis of intestinal cells shown in Supplementary Figure 6, it looks as if there may be decreases in total CD45+ cells and MHCII+ macrophages in the FIRE-/- mice. Were these changes significant?

The % of MHCII+ macrophages is not significantly different in the SI ($p=0.4003$ via a two-tailed t-test) or the colon ($p=0.0995$). We have included more data more points for the colon for %CD45+ cells ($p=0.0827$) and updated Supplementary Figure 6 and the figure legend accordingly.